# Generative Models for Long Time Series: Approximately Equivariant Recurrent Network Structures for an Adjusted Training Scheme

## Abstract

We present a simple yet effective generative model for time series data based on a Variational Autoencoder (VAE) with recurrent layers, referred to as the Recurrent Variational Autoencoder with Subsequent Training (RVAE-ST). Our method introduces an adapted training scheme that progressively increases the sequence length, addressing the challenge recurrent layers typically face when modeling long sequences. By leveraging the recurrent architecture, the model maintains a constant number of parameters regardless of sequence length. This design encourages approximate time-shift equivariance and enables efficient modeling of long-range temporal dependencies. Rather than introducing a fundamentally new architecture, we show that a carefully composed combination of known components can match or outperform state-of-the-art generative models on several benchmark datasets. Our model performs particularly well on time series that exhibit quasi-periodic structure, while remaining competitive on datasets with more irregular or partially non-stationary behavior. We evaluate its performance using ELBO, Fréchet Distance, discriminative scores, and visualizations of the learned embeddings.

## 1 Introduction

Time series data, particularly sensor data, plays a crucial role in science, industry, energy, and health. With the increasing digitization of companies and other institutions, the demand for advanced methods to handle and analyze time series sensor data continues to grow. Sensor data often exhibits distinct characteristics: it is frequently multivariate, capturing several measurements simultaneously, and may involve high temporal resolutions, where certain anomalies or patterns of interest only become detectable in sufficiently long sequences. Furthermore, such data commonly displays approximate stationarity or quasi-periodic behavior, reflecting repetitive patterns influenced by the underlying processes. These unique properties present both opportunities and challenges in the development of methods for efficient data synthesis and analysis, which are essential for a wide range of applications. Time series data analysis spans tasks such as forecasting (Siami-Namini et al., 2019), imputation (Tashiro et al., 2021; Luo et al., 2018), anomaly detection (Hammerbacher et al., 2021), and data generation. Of these, data generation stands out as the most general task, as advances in generative methods often yield improvements across the entire spectrum of time series applications (Murphy, 2022).

Recurrent neural networks, particularly Long Short-Term Memory (LSTM) networks (Hochreiter & Schmidhuber, 1997), are well-known for their ability to model temporal dynamics and capture dependencies in sequential data. However, their effectiveness tends to diminish with increasing sequence length, as maintaining long-term dependencies can become challenging (Zhu et al., 2023) where in contrast, convolutional neural networks (CNNs) (LeCun et al., 1998) demonstrate superior scalability for longer sequences (Bai et al., 2018). For instance, TimeGAN (Yoon et al., 2019) represents a state-of-the-art approach for generating synthetic time series data, particularly effective for short sequence lengths. In its original paper, TimeGAN demonstrates its capabilities on samples with sequence lengths of $l = 24$, showcasing limitations of LSTM-based architectures. By contrast, a model like WaveGAN (Donahue et al., 2019), which is built on

a convolutional architecture, is trained on significantly longer sequence lengths, with $l = 16384$ at minimum. This contrast highlights the fundamental differences and capabilities between recurrent and convolutional networks.

The limitations of LSTMs in modeling long-term dependencies are not restricted to time series data but also impact their performance in other domains, such as natural language processing (NLP). Early applications of attention mechanisms integrated with recurrent neural networks like LSTMs (Bahdanau, 2014) have largely been replaced by Transformer architectures (Vaswani et al., 2017), which excel in data-rich tasks due to their parallel processing capabilities and expressive attention mechanisms. While Transformer architectures have shown exceptional results in NLP (Radford et al., 2019), their application to time series data remains challenging. This is due in part to the self-attention mechanism's quadratic scaling in memory and computation with sequence length (Katharopoulos et al., 2020), which makes them less practical for very long sequences. Additionally, the inductive bias of Transformers differs from that of recurrent models: Transformers rely on positional encodings to model temporal structure, whereas recurrent architectures such as LSTMs process data sequentially by design, which inherently embeds a sense of temporal order into the model dynamics. This sequential processing makes recurrent models particularly well-suited for long, approximately stationary time series, where preserving temporal continuity over extended horizons can be highly beneficial.

Among the primary approaches for generative modeling of time series, three dominant frameworks have emerged: Generative Adversarial Networks (GANs) (Goodfellow et al., 2020), Variational Autoencoders (VAEs) (Kingma & Welling, 2014; Fabius & Van Amersfoort, 2014), and, more recently, Diffusion Models (Ho et al., 2020). Diffusion Models have demonstrated impressive capabilities in modeling complex data distributions, but their significant computational demands, high latency, and complexity make them less practical for many applications (Yang et al., 2024). Moreover, in terms of practical applications, there are often constraints in both time and computational resources, which limit the feasibility of performing extensive fine-tuning for each individual dataset. A general, well-performing approach that is both simple and efficient is therefore more desirable. In this context, VAEs stand out for their simplicity and direct approach to probabilistic modeling. In our work, we focus on VAEs and propose a novel method for training VAEs with recurrent layers to handle longer sequence lengths. We argue that VAEs are particularly suited for generation of time series data, as they explicitly learn the underlying data distribution, making them robust, interpretable, and straightforward to implement.

Our major contributions are:

- We introduce a novel combination of inductive biases, network topology, and training scheme in a recurrent variational autoencoder architecture. Our model integrates approximate time-translation equivariance into a recurrent structure, introducing a strong inductive bias toward stationary time series. Unlike existing recurrent or convolutional generative models, our architecture maintains a fixed number of parameters, independent of the sequence length.

- We propose a simple yet effective training procedure—Recurrent Variational Autoencoder Subsequent Train (RVAE-ST)—that progressively increases the sequence length during training. This scheme leverages the model's sequence-length-invariant parameterization and mitigates the typical limitations of recurrent layers in capturing long-range dependencies. It is particularly suited for approximately stationary datasets and contributes significantly to our model's performance.

- We conduct extensive experiments on five benchmark datasets and compare our method against a broad range of strong baselines, including models based on GANs, VAEs, convolutions, diffusion processes, and Transformers. This diverse set covers the most prominent architectural families in time-series generation and ensures a fair and comprehensive evaluation.

- To evaluate generative quality, we employ a comprehensive set of evaluation metrics, including ELBO (Evidence Lower Bound) of simpler generative models. Discriminative Score, and visualizations via PCA and t-SNE. We additionally apply the Contextualized Fréchet Inception Distance (Context FID) to better capture the alignment between global structure and local dynamics.

Our implementation, including preprocessing and model training scripts, is available at `https://github.com/ruwenflk/rvae-st`.

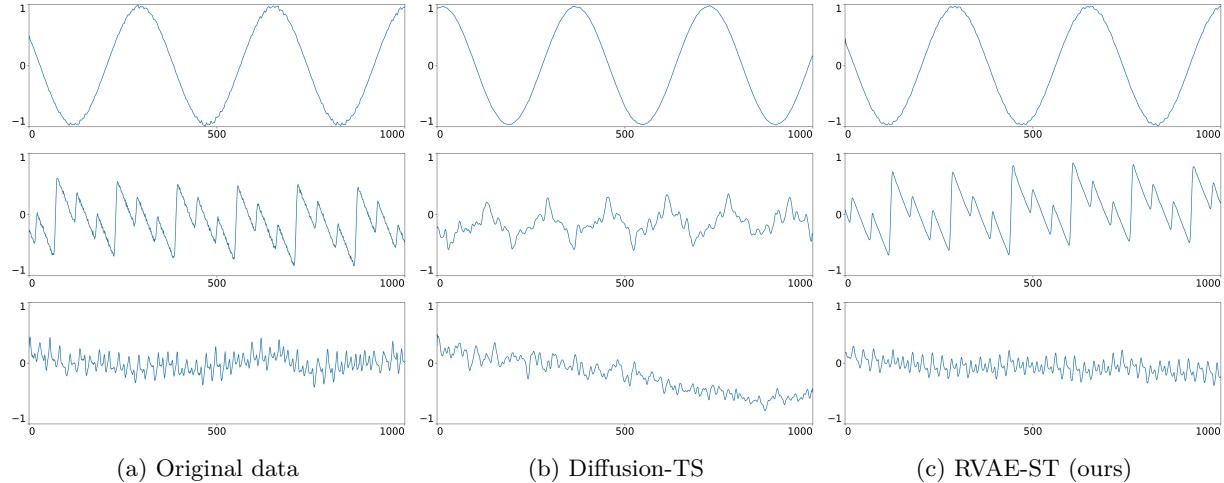

(a) Original data        (b) Diffusion-TS        (c) RVAE-ST (ours)

Figure 1: This figure shows three excerpts from samples of the electric motor dataset (5.1), each with a sequence length of $l = 1000$. Sample (a) is taken from the original dataset. Sample (b) is generated using Diffusion-TS (Yuan & Qiao, 2024), a transformer-based state-of-the-art approach in time series generation. Sample (c) is generated using our model, trained with the proposed subsequent training scheme. The first row in the figure displays the voltage of one of the phases. In the original sample (a), the extremities of the voltage waveform exhibit pronounced volatility, particularly at peak and trough points. This characteristic remains clearly visible in the output of model (c), whereas it is notably reduced in model (b). The second row shows the DC-bus voltage. The signal is characterized by a distinctive sawtooth-like pattern, where three gradual drops are each followed by an abrupt upward jump. Model (c) reproduces this pattern well, although the waveform appears slightly smoothed compared to the original. Model (b) captures the general frequency of the signal but fails to replicate the sawtooth-like structure. The third row shows the effective motor current in the fixed coordinates of the stator. This channel exhibits both a high-frequency component, which gives the signal a noisy appearance, and a low-frequency oscillation reflecting the long-term behavior. Model (c) closely resembles the original (a), capturing both components. Model (b) approximates the low-frequency trend but deviates significantly in the high-frequency range.

## 2 Prerequisites

### 2.1 Variational Autoencoder

Given the input dataset X, the goal is to find a probability density $p_\theta$ with high marginal likelihood (or evidence) $p_\theta(x)$ for $x \in X$. By introducing $z = z_{1:m}$ latent variables and assuming the joint density $p_\theta(x, z) = p_\theta(z)p_\theta(x|z)$, we get the intractable integral $p_\theta(x) = \int p_\theta(z)p_\theta(x|z)dz$. The evidence $p_\theta(x)$ can be approximated by computing the evidence lower bound or ELBO

$$\mathcal{L}_{\theta,\phi}(x) = \mathbb{E}_{q_\phi(z|x)}[\log p_\theta(x|z)] + D_{\mathbb{KL}}(q_\phi(z|x) \,||\, p_\theta(z)) = \mathcal{L}_E + \mathcal{L}_R \leq \log p_\theta(x).$$

$\mathcal{L}_E$ is the reconstruction loss (negative log likelihood) and $\mathcal{L}_R$ is the KL-Divergence loss (Murphy, 2022). $p_\theta(z)$ is the prior distribution which is usually set to $\mathcal{N}(0, I)$. The ELBO $\mathcal{L}_{\theta,\phi}(x) \leq \log p_\theta(x)$ is a lower bound to the log marginal likelihood and is maximized by training a VAE.

A VAE is a generative model that maps a sample $x \in X$ into a probability distribution in latent space $z$ by using an inference network $q_\phi(z|x) = \mathcal{N}(z|\mu, \text{diag}(\exp(\log(\sigma))))$ with $(\mu, \log(\sigma)) = e_\phi(x)$ where $e_\phi(x)$ is the encoder. The latent variable $z = \mu + \sigma \odot \epsilon$ where $\odot$ is the entrywise multiplication with $\epsilon = \mathcal{N}(0, I)$ is then passed through the generator model or decoder network $p_\theta(x|z)$ giving us a reconstructed sample $\widetilde{x}$.

Generating new timeseries samples works by just taking a sample from the known prior distribution $p_\theta(z)$ and feeding it through the generative model $p_\theta(x|z)$.

## 3 Related Work

### 3.1 Deep Generative Models for Time Series

Time-series generation has been explored across various deep generative paradigms, including GANs, VAEs, Transformers, and diffusion models. Early approaches focused on recurrent structures: C-RNN-GAN ((Mogren, 2016)) used LSTM-based generators and discriminators, while RCGAN(Esteban et al., 2017) introduced label-conditioning for medical time series. TimeGAN(Yoon et al., 2019) combined adversarial training, supervised learning, and a temporal embedding module to better capture temporal dynamics. Around the same time, WaveGAN(Donahue et al., 2019) introduced a convolutional GAN architecture for raw audio synthesis, illustrating that convolutional models can also be effective for generative tasks in the time domain. TimeVAE(Desai et al., 2021b) further explored this direction by proposing a convolutional variational autoencoder tailored to time-series data. PSA-GAN (Paul et al., 2022) employed progressive growing (Karras et al., 2018), incrementally increasing temporal resolution during training by adding blocks composed of convolution and residual self-attention to both the generator and discriminator. This fundamentally differs from our approach, which extends sequence length rather than resolution.

Recent advances in time-series generation have explored diffusion-based and hybrid Transformer architectures. Diffusion-TS (Yuan & Qiao, 2024) introduces a denoising diffusion probabilistic model (DDPM) tailored for multivariate time series generation. It employs an encoder-decoder Transformer architecture with disentangled temporal representations, incorporating trend and seasonal components through interpretable layers. Unlike traditional DDPMs, Diffusion-TS reconstructs the sample directly at each diffusion step and integrates a Fourier-based loss term. Time-Transformer (Liu et al., 2024) presents a hybrid architecture combining Temporal Convolutional Networks (TCNs) and Transformers in a parallel design to simultaneously capture local and global features. A bidirectional cross-attention mechanism fuses these features within an adversarial autoencoder framework (Makhzani et al., 2016). This design aims to improve the quality of generated time series by effectively modeling complex temporal dependencies.

A common limitation across all these approaches is their focus on relatively short sequence lengths. Many models, including TimeGAN, TimeVAE, and Time-Transformer, are evaluated at $l = 24$. Only Diffusion-TS and PSA-GAN extend this slightly, with ablations up to $l = 256$, leaving the performance on significantly longer sequences largely unexplored.

### 3.2 Recurrent Variational Autoencoders

The Recurrent Variational Autoencoder (RVAE) was introduced by Fabius & Van Amersfoort (2014), combining variational inference with basic RNNs for sequence modeling. In this architecture, the latent space is connected to the decoder via a linear layer, and the sequence is reconstructed by applying a sigmoid activation to each RNN hidden state.[1] We build on this framework by replacing the basic RNNs with LSTMs (or GRUs) and using a repeat-vector mechanism that injects the same latent vector at every time step of the decoder. This design encourages the latent code to encode global sequence properties, while the LSTM handles temporal dependencies. Instead of a sigmoid, we apply a time-distributed linear layer, preserving approximate time-translation equivariance (see Section 4.1).

Unlike dynamic VAEs (dVAE) that use a sequence of latent variables to increase flexibility (Girin et al., 2021), we opt for a single latent vector of fixed size across the entire sequence. This choice reflects our focus on the inductive bias of translational equivariance and stationarity, where the latent code is meant to capture global properties of the sequence while allowing the decoder to model local temporal dynamics. This distinction means that, unlike in dVAE models, the latent code does not change over time, aligning with the assumptions of our model and the goal of preserving global structure while modeling temporal relationships.

---

[1] https://github.com/arunesh-mittal/VariationalRecurrentAutoEncoder/blob/master/vrae.py

While recurrent models in VAEs are often susceptible to posterior collapse (He et al., 2019), we do not view this phenomenon as an issue but rather as a natural feature of our model's design. By relying on the equivariant structure, we encourage the latent space to focus on capturing global sequence properties, which aligns with our goal of modeling temporal dependencies while maintaining stationarity over time.

## 4 Methods

### 4.1 Equivariant Vec2Seq generator

The objective of our model development was to create a generator that incorporates an inductive bias toward time-shift invariance in its learned distribution $p_\theta(x)$ , while being capable of generating sequences of variable lengths from fixed-length inputs, as seen in vector-to-sequence (Vec2Seq) models. This design choice was motivated by the idea that such properties would improve model performance, particularly on (semi-)stationary and quasi-periodic time series.

Assuming that all variables represent probabilities rather than samples, we define an approximately time-shift invariant $p_\theta(x)$ for time dependent data $x : \mathbb{Z} \to \mathbb{R}^c : t \mapsto x(t)$ as follows:

$$\int p_\theta(x|z)p_\theta(z)\,dz = p_\theta(x) \approx p_\theta(\tau(x)) = \int p_\theta(\tau(x)|z)p_\theta(z)\,dz, \tag{1}$$

where $\tau$ is the time-shift operator $\tau : (\mathbb{R}^c)^{\mathbb{Z}} \to (\mathbb{R}^c)^{\mathbb{Z}} : x(t) \mapsto x(t+1)$. This definition says that each sample can be generated with approximately same probability starting at each time step. This holds when $p_\theta(x|z) \approx p_\theta(\tau(x)|z)$, which implies that a generator model with an inductive bias toward time-shift invariance can be achieved using a $\tau$-equivariant network topology. In practice, we compute probabilities on finite time windows of $x$ such as $x_{|\{1,\ldots,\ell\}} : \{1,\ldots,\ell\} \to \mathbb{R}^c$ and by abuse of notation write $x \in \mathbb{R}^{\ell \times c}$. Building on this idea, RNNs process data sequentially and reuse the same transition function at each time step, which results in approximate temporal shift equivariance. However, due to hidden state initialization and finite context (e.g., truncation or saturation effects), true equivariance does not strictly hold. For example, the hidden state at time $t$ may still implicitly depend on absolute position (especially at the beginning of a sequence). To illustrate this further, consider the case where we use an LSTM cell $(y_i, h_{i+1}, c_{i+1}) = f(x_i, h_i, c_i)$ mapping input $x_i$, hidden state $h_i$, and cell state $c_i$ to the output $y_i$. We will denote the map producing the hidden and cell state by $(h_{i+1}, c_{i+1}) = \hat{f}(x_i, h_i, c_i)$. Now consider mapping two overlapping time series $X = [x_0, \ldots, x_{n-1}]$ and $X' = [x_1, \ldots, x_n]$ via this LSTM-cell. Of course, in general $f(x_0, h, c) \neq f(x_1, h, c)$ for initializations $h$ and $c$ of hidden and cell state. However, for long time series (i.e. $n \gg 0$) we can usually expect convergence in hidden and cell state over time, more precisely

$$\hat{f}(x_k, \hat{f}(x_{k-1}, \hat{f}(x_{k-2}, \hat{f}(\ldots, \hat{f}(x_0, h, c)\ldots)))) \approx \hat{f}(x_k, \hat{f}(x_{k-1}, \hat{f}(x_{k-2}, \hat{f}(\ldots, \hat{f}(x_1, h, c)\ldots)))), \tag{2}$$

i.e. shifting a long time series by will be approximately equivariant.

Given that our focus is on time series data with quasi-stationary behavior, other architectures like convolutional layers and transformers face specific limitations. Convolutional layers are commonly employed for building equivariant network structures due to their ability to recognize patterns in data regardless of their position within the input. This property makes them highly effective, especially in image processing. However, in the context of time series, convolutional layers alone are insufficient for generating sequences of variable lengths from fixed-length inputs. The process of upscaling in time series typically enhances the resolution of the data within a fixed time window, similar to image processing, rather than extending the sequence itself. (Paul et al., 2022) is an example for this. This fundamental distinction makes convolutional layers less suitable for handling time series with variable-length sequences. Transformers, while powerful for capturing long-range dependencies, face challenges when applied to time series data. One of the main drawbacks is the quadratic increase in computational complexity as sequence length grows (Katharopoulos et al., 2020) due to the self-attention mechanism. This makes them less efficient for handling long time series compared to recurrent architectures. We further note that transformers are not translation equivariant but

permutation equivariant. While transformers are permutation equivariant and can handle inputs in any order, they do not capture the inherent sense of temporal order in time series data. This sense of order is especially important for modeling sequential dependencies in approximately stationary data, as considered in this paper, where time-shift equivariance is crucial. In contrast, LSTMs naturally embed temporal order into their model dynamics, enabling them to capture long-range temporal dependencies while maintaining an internal state.

## 4.2   RVAE-ST

The inference network $q_\phi(z|x)$ is implemented using stacked LSTM layers. Given the final point in time of a sequence, the output of the last LSTM layer is passed through two linear layers to determine $\mu$ and $\log(\sigma)$, which are then used to sample the latent variable $z$. Next, the generative network $p_\theta(x|z)$ reconstructs the data from the latent variable $z$.To achieve this, the latent variable $z$ is repeated across all time steps (using a repeat vector), ensuring that $z$ remains constant at each time step and is shared throughout the entire sequence. Mathematically, this can be expressed as:

$$z_t = z \quad \text{for all } t \in \{1, 2, \ldots, n\}$$

where $n$ denotes the total number of time steps in the sequence. The repeat vector is followed by stacked LSTM layers. Finally, a time-distributed linear layer is applied in the output. This layer operates independently at each time step, applying the same linear transformation to the LSTM output at every time step, which can be viewed as a $1 \times 1$ convolution across the time dimension, with shared weights across all time steps.

The time-distributed layer is inherently equivariant with respect to time-translation, preserving temporal structure and shifts over time. Together with our LSTM-based approach and the repeat-vector mechanism, this design ensures that the number of trainable parameters remains independent of the sequence length, while also enabling an adapted training scheme that can accommodate increasing sequence lengths. Details and hyperparameters are provided in Appendix A.2.

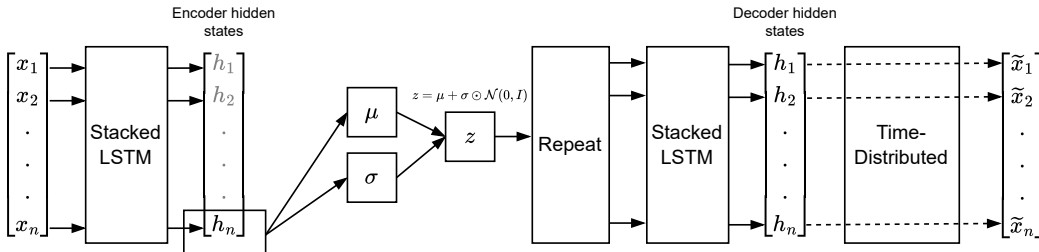

Figure 2: This figure illustrates the architecture of our model. Both the encoder and decoder are based on stacked LSTM layers. The encoder's final hidden states, denoted as $h_n$, are used to compute the parameters $\mu$ and $\log(\sigma)$, from which the latent variable $z$ is sampled. The latent variable $z$ is then repeated across all time steps and used as the input to the decoder. The decoder generates the sequence step-by-step, with each individual output passed through a time-distributed linear layer. This time-distributed layer applies the same linear transformation at each time step to the LSTM states, ensuring parameter sharing across the entire sequence during this transformation. Throughout the network, approximate equivariance is maintained with respect to time translation, and the number of trainable parameters remains constant regardless of the sequence length.

## 4.3   Training scheme for sequence lengths

Training a recurrent neural network such as an LSTM to produce consistent long time sequences is challenging, as recurrent layers have a limited capacity to preserve information over extended temporal ranges.

Here, we suggest a training scheme that allows training over longer time sequences for our RVAE-ST model, which is visualized in figure 2.

We begin by splitting the dataset into smaller chunks, as is commonly done. The model is initially trained on short sequences, which ensures stable training and facilitates faster progress. After training on short sequences, we progressively increase the sequence length, rebuild the dataset by splitting it into new, longer chunks, and continue training on the longer sequences. Each training phase is completed when no improvement is observed in the validation loss for a predefined number of epochs. This process is repeated as the sequence length increases. This method stabilizes the training process for long time sequences and improves the final results, as demonstrated in our experiments.

We motivate this probabilistically for a time series $x$ of length $l$, hidden features distributed over time $h$ again of length $l$, and a fixed latent vector $z$, where we have a recurrent structure over the $h$ via

$$p(x,h,z) = p(z) \prod_{i=1}^{l} p(h_i|z, h_{i-1}, \ldots, h_1) \cdot p(x_i|h_i).$$

The generative model $p(x,h|z)$ can now be approximated by only looking $t$ steps back in time.

$$
\begin{aligned}
p(x,h|z) &= \prod_{i=1}^{l} p(h_i|z, h_{i-1}, \ldots, h_1) \cdot p(x_i|h_i) \\
&\approx \prod_{i=1}^{l} p(h_i|z, h_{i-1}, \ldots, h_{\max(1,i-t)}) \cdot p(x_i|h_i)
\end{aligned}
\tag{3}
$$

Hence, training to generate shorter sequences yields an approximation of generating long sequences. We do not provide general recommendations for which initial sequence length or increment values work best, as the sequence lengths chosen in our experiments are somewhat arbitrary and may vary depending on the dataset.

## 5 Experiments

In our experiments, we compare the performance of RVAE-ST to comparison models. We emphasize that, to ensure better comparability, we did not perform extensive hyperparameter tuning in our experiments. In all experiments, including different datasets and varying sequence lengths, we used exactly the same hyperparameters on the model. For the training procedure, we started with a sequence length of 100 and progressively increased it by 100 in each subsequent training phase, until reaching a maximum sequence length of 1000. In our experiments, we compare the performance of the models at sequence lengths of 100, 300, 500, and 1000.

To evaluate performance, we employ a combination of short-term consistency measures based on independently generated ELBOs, the discriminative score, and the contextual FID score. Additionally, we perform visual comparisons between the training and generated data distributions using dimensionality reduction techniques such as PCA and t-SNE. All reported results were tested for statistical significance using the Wilcoxon rank-sum test (Wilcoxon, 1992). In cases where the difference was not statistically significant, multiple values are highlighted in bold.

### 5.1 Data Sets

For our experiments we use three multivariate sensor datasets with it's typical semi-stationary behavior. We specifically selected datasets with a minimum size, as this is necessary for training generative models effectively, while still ensuring adequate diversity and the ability to robustly capture underlying patterns and structures.

**Electric motor (EM)(Wißbrock & Müller, 2025; Mueller, 2024):** This dataset was collected from a three-phase motor operating under constant speed and load conditions, with different combinations stored in separate files. We use only the file `H1.5`, selected arbitrarily among them. It exhibits periodic behavior with

similar, repeating patterns. The data was recorded at a sampling rate of 16,kHz. Out of the twelve initially available channels, four were removed due to discrete behavior or abrupt changes, leaving only smooth, continuous signals suitable for learning. The resulting dataset contains approximately 250,000 datapoints and represents the most stationary real-world dataset used in our experiments.

**Ecg data (ECG)**[2] **Goldberger et al. (2000):** This dataset contains a two-channel electrocardiogram recording from the MIT-BIH Long-Term ECG Database. It has nearly 10 million time steps of which we use the first 500,000 for training. The ECG signals are nearly periodic but exhibits variations in frequency and occasional irregular arrhythmias, making the dataset slightly less stationary than the EM dataset. While ECG data serves as a suitable example in generating long sequences, our objective is not to produce medically usable data. We acknowledge that specialized models are likely more appropriate for medical applications, e.g. (Neifar et al., 2023).

**ETTm2 (ETT)**[3]**:** The ETTm2 dataset, collected between 2016 and 2018, consist of sensor measurements such as load and oil temperature from electricity transformers. These datasets contain short-term periodical patterns, long-term periodical patterns, long-term trends, and various irregular patterns. Out of the three datasets, it is the smallest, containing 69680 datapoints. It was published in the context of time series forecasting Zhou et al. (2021a) and is a widespread used dataset (Zhou et al., 2021b; Zhang et al., 2024; Zhu et al., 2023). Out of the three in this paper, is the least stationary dataset.

**Synthetic Sine:** The sine dataset is generated by sampling 5 independent sine waves, each with randomly chosen frequencies and phases, which are drawn independently from uniform distributions in the range $[0, 0.1]$. This dataset is highly stationary and noise-free, with each channel following a smooth, periodic sine wave. It is commonly used as a standard benchmark in time-series modeling tasks (Yoon et al., 2019; Desai et al., 2021b; Yuan & Qiao, 2024).

**MetroPT3 (Davari et al. (2021)):** The MetroPT3 dataset is used for predictive maintenance, anomaly detection, and remaining useful life (RUL) prediction in compressors. It consists of multivariate time-series data from several analogue and digital sensors installed on a compressor, including signals such as pressures, motor current, oil temperature, and electrical signals from air intake valves. The data were logged at a frequency of 1Hz. Similar to the Electric Motor dataset, we removed non-continuous or discrete signals, leaving only smooth, continuous signals suitable for learning. Out of the original 1.5 million time steps, we only used the first 500,000 for our experiments. While the dataset contains recurring patterns, it is one of the less stationary datasets in our study, as the frequency of these patterns exhibits significant variance, and some signals occasionally drop out completely.

### 5.2 Comparison Models

In this subsection, we describe the baseline models selected for comparison in our experiments. These models are chosen for their relevance to time series generation and their established use in similar contexts.

**TimeGAN**(Yoon et al., 2019)**:** A GAN-based model that is considered state-of-the-art in generation of times series data. TimeGAN's generator has a recurrent structure like RVAE-ST. A key difference is that it's latent dimension is equal to the sequence length. Notably, equivariance on this model is lost on the output layer of the generator which maps all hidden states at once through a linear layer to a sequence. On its initial paper release, TimeGAN was tested and compared to other models on a small sequence length of $l = 24$.

**WaveGAN** (Donahue et al., 2019)**:** A GAN-based model developed for generation of raw audio waveforms. WaveGAN's generator is based on convolutional layers. It doesn't rely on typical audio processing techniques like spectrogram representations and is instead directly working in the time domain, making it also suitable for learning time series data. It is designed to exclusively support sequence lengths in powers of 2, specifically $2^{14}$ to $2^{16}$. Notably, WaveGAN loses it's equivariance on a dense layer between the latent dimension and the generator, however the generator itself completely maintains equivariance with its upscaling approach. In our experiments, it was trained with the lowest possible sequence length of $2^{14}$, and the generated samples

---

[2]https://physionet.org/content/ltdb/1.0.0/14157.dat
[3]https://github.com/zhouhaoyi/ETDataset

were subsequently split to match the required sequence length. In (Yoon et al., 2019), WaveGAN was outperformed by TimeGAN on low sequence length.

**TimeVAE** (Desai et al., 2021b)**:** A VAE-based model designed for time series generation using convolutional layers. Analogous to WaveGAN, it loses equivariance between the latent dimension and the decoder and additionally it loses equivariance on the output layer where a flattened convolutional output is passed through a linare layer. It has demonstrated performance comparable to that of TimeGAN.

**Diffusion-TS** (Yuan & Qiao, 2024): A generative model for time series based on the diffusion process framework. It combines trend and seasonal decomposition with a Transformer-based architecture. A Fourier basis is used to model seasonal components, while a low-degree polynomial models trends. Samples are generated by reversing a learned noise-injection process. While the model leverages the global structure of sequences, it lacks time-translation equivariance: this is due both to the use of position embeddings in the Transformer component and to the fixed basis decomposition, which breaks shift-invariance.

**Time-Transformer** (Liu et al., 2024): An adversarial autoencoder (AAE) model tailored for time series generation, integrating a novel Time-Transformer module within its decoder. The Time-Transformer employs a layer-wise parallel design, combining Temporal Convolutional Networks (TCNs) for local feature extraction and Transformers for capturing global dependencies. A bidirectional cross-attention mechanism facilitates effective fusion of local and global features. While TCNs are inherently translation-equivariant, this property is overridden by the Transformer's position encoding and attention structure, making the overall model not equivariant.

## 5.3   Evaluation by Context-FID Score

To evaluate the distributional similarity between real and generated time series, we use the Context-FID score (Paul et al., 2022), a variant of the Fréchet Inception Distance (FID) commonly used in image generation. In this adaptation, the original Inception network is replaced by TS2Vec (Yue et al., 2022), a self-supervised representation learning method for time series. The score is computed by encoding both real and generated sequences with a pretrained TS2Vec model and calculating the Fréchet distance between the resulting feature distributions. Lower scores indicate that the synthetic data better matches the distribution of the real data.

Across the different sequence lengths, RVAE-ST consistently outperforms all comparison models on the Electric Motor, ECG, and Sine datasets starting from $l = 300$. These datasets exhibit high stationarity, which aligns well with the inductive biases of our approach. On the less stationary MetroPT3 and ETT datasets, our model remains competitive, with TimeVAE surpassing it at $l = 1000$ for both datasets. Additionally, for MetroPT3, Diffusion-TS outperforms our model at $l = 500$.

## 5.4   Evaluation by Average ELBO

Next we evaluate the average Evidence Lower Bound (ELBO) on a synthetic dataset $\tilde{X} \in \mathbb{R}^{n_s \times l \times c}$ where $n_s$ represents the numbers of samples, $l$ denotes the sequence length, and $c$ the number of channels. We refer to this metric as $\mathcal{E}_{\text{avg}}(\tilde{X})$. In detail, we first train a VAE model on shorter sequence lengths $\ell \ll l$, which facilitates easier training. We denote it as the *ELBO model* $\tilde{\mathcal{L}}_{\theta,\phi} : \mathbb{R}^{\ell \times c} \to \mathbb{R}$.

We then calculate the *average ELBO*:

$$\mathcal{E}_{\text{avg}}(\tilde{X}) = \frac{1}{n_s(l-\ell)} \sum_{i=0}^{n_s-1} \sum_{t=0}^{l-\ell-1} \text{ELBO}_{\text{norm}} \left( \tilde{\mathcal{L}}_{\theta,\phi}(\tilde{X}_{i,t:t+\ell,\cdot}) \right), \tag{4}$$

where $\text{ELBO}_{\text{norm}} = \text{ELBO} \cdot (ct)^{-1}$ is a normalized ELBO, as explained in Appendix A.3. By normalizing the ELBO, we get a fairer comparison of datasets with different dimensionalities and varying sequence lengths.

$\mathcal{E}_{\text{avg}}(\tilde{X})$ gives us information about short term consistency over the whole synthetic dataset. We chose $\ell = 50$ which is half of the lowest sequence length in the experiments. A well trained *ELBO model* (An & Cho, 2015) allows us to evaluate the (relative) short term consistency of synthetic data in high accuracy

Table 1: **FID score** of synthetic time series for six models (see 5.2), computed on the five datasets (see 5.1) at sequence lengths $l = 100$, $l = 300$, $l = 500$, and $l = 1000$. Lower scores indicate better performance. Each score is based on 5000 generated samples, each evaluated 15 times, and reported with 1-sigma confidence intervals. RVAE-ST consistently outperforms all baselines on the highly stationary Electric Motor, ECG, and Sine datasets starting from $l = 300$. On the less stationary MetroPT3 and ETT datasets, performance is more competitive, with TimeVAE and Diffusion-TS outperforming our model at certain sequence lengths.

| Dataset | Model | Sequence lengths | | | |
| | | 100 | 300 | 500 | 1000 |
|---------|-------|------|------|------|------|
| Electric Motor | **RVAE-ST (ours)** | 0.35±0.04 | **0.12±0.01** | **0.10±0.01** | **0.24±0.02** |
| | TimeGAN | 1.03±0.07 | 3.77±0.30 | 3.07±0.24 | 33.7±1.69 |
| | WaveGAN | 0.55±0.04 | 0.75±0.07 | 0.87±0.14 | 1.41±0.24 |
| | TimeVAE | 0.16±0.01 | 0.97±0.11 | 1.06±0.14 | 1.19±0.09 |
| | Diffusion-TS | **0.04±0.00** | 0.69±0.06 | 1.10±0.11 | 1.93±0.13 |
| | Time-Transformer | 2.19±0.16 | 45.4±1.57 | 44.5±2.67 | 65.7±2.86 |
| ECG | **RVAE-ST (ours)** | **0.08±0.01** | **0.09±0.02** | **0.14±0.02** | **0.46±0.06** |
| | TimeGAN | 26.8±6.89 | 48.0±6.26 | 47.2±5.91 | 34.0±3.43 |
| | WaveGAN | 1.54±0.19 | 1.56±0.14 | 1.54±0.13 | 1.51±0.16 |
| | TimeVAE | 0.26±0.02 | 0.89±0.07 | 1.07±0.10 | 1.30±0.08 |
| | Diffusion-TS | 0.16±0.01 | 0.28±0.03 | 0.52±0.03 | 3.74±0.22 |
| | Time-Transformer | 1.34±0.11 | 29.7±1.78 | 33.0±2.28 | 40.3±2.44 |
| ETT | **RVAE-ST (ours)** | **0.58±0.05** | **0.65±0.07** | **0.79±0.07** | 1.82±0.16 |
| | TimeGAN | 1.51±0.19 | 5.76±0.43 | 13.7±1.28 | 17.7±1.57 |
| | WaveGAN | 3.49±0.22 | 3.90±0.37 | 4.38±0.39 | 4.94±0.42 |
| | TimeVAE | 0.66±0.08 | 0.72±0.08 | 0.97±0.10 | **1.56±0.14** |
| | Diffusion-TS | 0.90±0.11 | 1.18±0.18 | 2.16±0.17 | 2.55±0.27 |
| | Time-Transformer | 1.28±0.14 | 20.1±1.22 | 22.1±1.96 | 47.9±5.28 |
| Sine | **RVAE-ST (ours)** | 0.33±0.04 | **0.34±0.02** | **0.46±0.03** | **0.42±0.03** |
| | TimeGAN | 7.70±0.32 | 6.01±0.34 | 7.96±0.37 | 21.8±1.25 |
| | WaveGAN | 1.87±0.10 | 2.09±0.13 | 2.81±0.22 | 3.36±0.27 |
| | TimeVAE | 0.24±0.02 | 0.55±0.05 | 1.26±0.14 | 3.03±1.00 |
| | Diffusion-TS | **0.06±0.00** | 1.52±0.13 | 0.74±0.04 | 2.66±0.20 |
| | Time-Transformer | 0.31±0.02 | 4.10±0.21 | 51.2±1.94 | 74.5±3.85 |
| MetroPT3 | **RVAE-ST (ours)** | **0.26±0.04** | **0.65±0.07** | 2.81±0.37 | 2.84±0.22 |
| | TimeGAN | 5.79±0.32 | 10.1±0.79 | 18.6±1.06 | 35.1±3.74 |
| | WaveGAN | 1.14±0.09 | 1.82±0.12 | 2.04±0.16 | 2.43±0.18 |
| | TimeVAE | 0.67±0.05 | 1.32±0.13 | 2.02±0.29 | **2.08±0.31** |
| | Diffusion-TS | 1.07±0.06 | 1.17±0.12 | **1.82±0.09** | 6.97±0.75 |
| | Time-Transformer | 2.28±0.24 | 5.25±0.46 | 22.9±1.45 | 352±66.1 |

and low variance. To ensure reliable assessment of sample quality, we prevented overfitting of the *ELBO model* by applying early stopping after 50 epochs without improvement and restoring the best weights. In our experiments, we employed two distinct *ELBO models* for calculating $\mathcal{E}_{\mathrm{avg}}(\tilde{X})$. The first model is based on the RVAE-ST architecture, while the second utilizes the TimeVAE framework (Desai et al., 2021a). The use of a TimeVAE-based *ELBO model* provides an additional evaluation to ensure that the RVAE-ST-based model is not biased toward our own generated samples. As detailed in Appendix A.6, the results obtained using TimeVAE are highly similar to those produced by the RVAE-ST-based model.

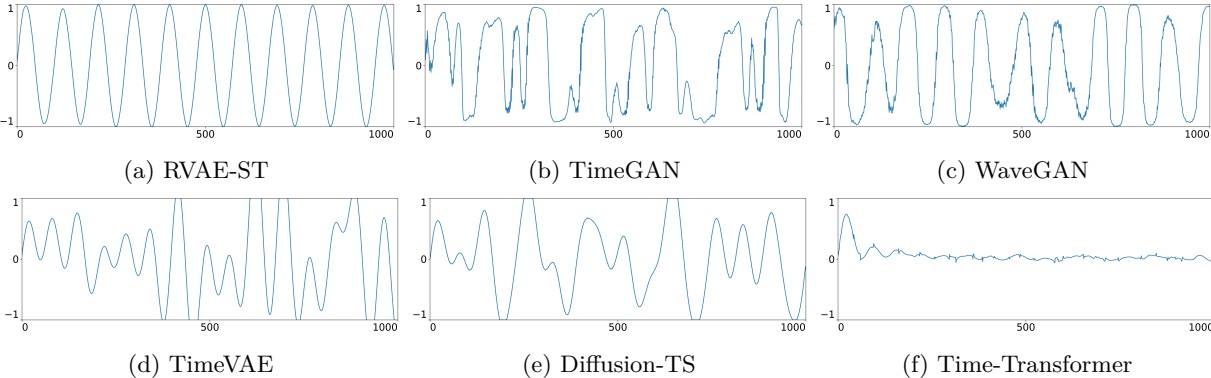

Figure 3: Representative samples for each model at sequence length $l = 1000$ on a stationary dataset. RVAE-ST is the only model capable of consistently generating correct sinusoidal curves, demonstrating its ability to capture the stationary nature of the data.

On the Electric Motor dataset, RVAE-ST consistently generates the best samples. Similarly, on the ETT dataset, RVAE-ST remains superior starting from $l = 300$, demonstrating its ability to generate high-quality samples even on less stationary datasets.

The Time-Transformer model exhibits specific behavior that warrants further examination. On the Sine and ECG datasets, it generates relatively flat samples that fail to capture the true characteristics of the datasets. Unfortunately, the ELBO score is unable to detect this issue in the generated samples, leading to an overestimation of the model's performance based on its ELBO score. When this issue is taken into account, RVAE-ST outperforms all other models on the ECG dataset starting at $l = 300$ and on the Sine dataset starting at $l = 500$. The Sine dataset, however, requires special consideration due to its unique characteristics. One key challenge is that the individual channels are not correlated. As a result, the inference and subsequent reconstruction by the ELBO model become dysfunctional, leading to excessively negative ELBO values. This effect is further enhanced when poor samples appear in at least one channel of the generated samples. Therefore, the ELBO score is only of limited significance for this dataset, and a visual evaluation of the generated samples is necessary to assess their quality effectively.

In Figure 3, representative samples for each model are shown for a sequence length of $l = 1000$. RVAE-ST is the only model that can generate proper and consistent sine curves, which are characteristic of the dataset. The Sine dataset, as a clear example of a stationary time series, further supports our hypothesis that a translation-equivariant network architecture is particularly effective at generating consistent, high-quality long-range sequences in such scenarios. On the MetroPT3 dataset, TimeVAE performs best at $l = 500$, while Diffusion-TS outperforms all other models at the other sequence lengths.

## 5.5 Evaluations by Discriminative Score

The discriminative score $\mathcal{D}$ was introduced by (Yoon et al., 2019) as a metric for quality evaluation of synthetic time series data. For the discriminative score a simple 2-layer RNN for binary classification is trained to distinguish between original and synthetic data. Implementation details are in the appendix A.5. It is defined as $\mathcal{D} = |0.5 - a|$, where $a$ represents the classification accuracy between the original test dataset and the synthetic test dataset that were not used during training. The best possible score of 0 means that the classification network cannot distinguish original from synthetic data, whereas the worst score of 0.5 means that the network can easily do so.

The discriminative score provides particularly meaningful insights when it allows for clear distinctions between models, which is best achieved by avoiding scenarios where the score consistently reaches its best or worst possible values across different models. To ensure consistency, we used the same fixed number of samples for training the discriminator across all experiments, regardless of sequence length. This fixed sample size was found to be suitable for our experimental setup.

Table 2: **Average *ELBO* score** $\mathcal{E}_{\mathrm{avg}}(\tilde{X})$ of synthetic time series for six models (see 5.2), computed on the five datasets (see 5.1) at sequence lengths $l = 100$, $l = 300$, $l = 500$, and $l = 1000$. Higher scores indicate better performance. Each score is based on 1500 generated samples evaluated with an *ELBO model* using the RVAE-ST architecture, with 1-sigma confidence intervals. Note that while the ELBO score is generally informative, it can overestimate quality on certain datasets such as Sine and ECG, where implausible outputs may go undetected. For the Sine dataset in particular, uncorrelated channels and high sensitivity to local artifacts limit the reliability of this metric.

| Dataset | Model | Sequence lengths | | | |
|---|---|---|---|---|---|
| | | **100** | **300** | **500** | **1000** |
| Electric Motor | **RVAE-ST(ours)** | **1.62±0.69** | **1.65±0.60** | **1.66±0.03** | **1.65±0.03** |
| | TimeGAN | 1.20±0.59 | 1.33±0.48 | 1.13±0.56 | -4.05±2.41 |
| | WaveGAN | 1.54±0.11 | 1.54±0.16 | 1.54±0.14 | 1.53±0.37 |
| | TimeVAE | 1.49±0.88 | 1.38±1.34 | 1.09±2.21 | 0.31±3.24 |
| | Diffusion-TS | 1.58±0.06 | 1.36±0.26 | 1.38±0.24 | 1.30±0.25 |
| | Time-Transformer | 0.98±2.46 | -28.9±3.33 | -21.7±0.91 | -28.4±4.12 |
| ECG | **RVAE-ST(ours)** | 1.64±0.13 | 1.64±0.18 | 1.63±0.20 | 1.59±0.27 |
| | TimeGAN | -14.6±1.87 | -14.6±1.41 | -13.7±6.67 | -15.3±2.57 |
| | WaveGAN | 1.12±0.81 | 1.11±0.87 | 1.10±0.86 | 1.10±0.83 |
| | TimeVAE | 1.55±0.37 | 1.37±0.65 | 1.26±0.70 | 0.87±0.92 |
| | Diffusion-TS | **1.65±0.07** | 1.64±0.19 | 1.60±0.29 | 1.29±1.00 |
| | Time-Transformer | 1.07±0.85 | **1.68±0.05** | **1.68±0.05** | **1.68±0.05** |
| ETT | **RVAE-ST(ours)** | 1.49±0.52 | **1.50±0.40** | **1.52±0.35** | **1.53±0.63** |
| | TimeGAN | 1.39±0.70 | 0.85±3.36 | -4.29±9.66 | -0.38±0.65 |
| | WaveGAN | 1.40±0.53 | 1.39±0.70 | 1.42±0.51 | 1.42±0.48 |
| | TimeVAE | 1.47±0.94 | 1.20±1.54 | 0.89±1.99 | 0.42±2.45 |
| | Diffusion-TS | **1.50±0.18** | 1.49±0.26 | 1.50±0.27 | 1.50±0.17 |
| | Time-Transformer | 1.07±1.93 | 1.38±0.86 | 1.49±0.14 | -39.9±5.84 |
| Sine | **RVAE-ST(ours)** | 1.09±0.51 | -4.96±6.00 | -5.01±5.82 | -5.13±5.73 |
| | TimeGAN | -3.39±3.84 | -9.38±6.87 | -11.2±6.43 | -12.2±8.65 |
| | WaveGAN | -2.71±2.79 | -7.69±5.71 | -7.65±5.57 | -7.66±5.54 |
| | TimeVAE | 0.81±0.73 | -8.79±7.81 | -11.1±7.39 | -14.4±9.49 |
| | Diffusion-TS | **1.26±0.01** | **-3.58±4.94** | -5.53±5.24 | -7.33±5.07 |
| | Time-Transformer | 0.93±0.54 | -5.25±4.53 | **0.84±0.84** | **1.06±0.62** |
| MetroPT3 | **RVAE-ST(ours)** | 1.41±1.74 | 0.76±3.49 | 0.57±3.78 | 0.60±3.75 |
| | TimeGAN | 1.25±1.38 | 0.61±4.39 | **1.46±1.36** | -11.1±18.2 |
| | WaveGAN | -1.71±4.85 | -1.62±4.90 | -1.64±4.83 | -1.68±4.91 |
| | TimeVAE | -0.07±3.96 | -2.06±5.91 | -5.64±7.29 | -9.03±7.38 |
| | Diffusion-TS | **1.63±0.92** | **1.43±2.21** | 1.36±2.53 | **0.77±3.50** |
| | Time-Transformer | -2.30±5.81 | -3.05±6.55 | -2.97±0.55 | -302±14.4 |

As shown in Table 3, the Discriminative Score yields a less clear-cut picture compared to other evaluation metrics. The Wilcoxon rank-sum test reveals that in several cases, performance differences between models are not statistically significant.

On the Electric Motor dataset, RVAE-ST achieves the best performance from $l = 300$ onwards. For the ECG dataset, RVAE-ST outperforms all other models at $l = 1000$, while for shorter sequence lengths, its performance is comparable to that of Diffusion-TS. On the ETT dataset, RVAE-ST, TimeVAE, and Diffusion-TSperform similarly well across all sequence lengths, with no statistically significant differences. The Sine

Table 3: **Discriminative score** of synthetic time series for six models (see 5.2), computed on the five datasets (see 5.1) at sequence lengths $l = 100$, $l = 300$, $l = 500$, and $l = 1000$. A lower score indicates better performance. Each score is based on 15 independent discriminator runs and reported with 1-sigma confidence intervals. RVAE-ST performs best on the Electric Motor dataset from $l = 300$ onward and significantly outperforms all models on ECG at $l = 1000$, while showing comparable performance to Diffusion-TS at shorter lengths. For the ETT and Sine datasets, multiple models perform similarly depending on the sequence length. On MetroPT3, RVAE-ST is best at $l = 100$, while Diffusion-TS dominates for longer sequences. In cases without statistically significant differences (Wilcoxon rank-sum test), multiple scores are highlighted in bold.

| Dataset | Model | Sequence lengths | | | |
| --- | --- | --- | --- | --- | --- |
| | | 100 | 300 | 500 | 1000 |
| Electric Motor (EM) | **RVAE-ST (ours)** | **.121**±**.021** | **.032**±**.018** | **.038**±**.018** | **.085**±**.015** |
| | TimeGAN | .338±.030 | .477±.018 | .486±.013 | .500±.000 |
| | WaveGAN | .352±.009 | .416±.009 | .425±.011 | .444±.011 |
| | TimeVAE | .268±.214 | .226±.176 | .185±.083 | .152±.047 |
| | Diffusion-TS | **.112**±**.056** | .327±.130 | .396±.085 | .434±.084 |
| | Time-Transformer | .334±.098 | .500±.000 | .500±.000 | .500±.000 |
| ECG | **RVAE-ST (ours)** | **.012**±**.011** | **.009**±**.008** | **.016**±**.014** | **.009**±**.010** |
| | TimeGAN | .466±.125 | .500±000 | .500±.000 | .500±000 |
| | WaveGAN | .306±.155 | .300±.201 | .402±.153 | .298±.217 |
| | TimeVAE | **.034**±**.066** | .058±.120 | .131±.181 | .153±.177 |
| | Diffusion-TS | **.007**±**.007** | **.016**±**.016** | **.010**±**.015** | .382±.145 |
| | Time-Transformer | .216±.107 | .500±.000 | .496±.014 | .499±.002 |
| ETT | RVAE-ST (ours) | .179±.034 | **.172**±**.105** | **.189**±**.049** | **.132**±**.147** |
| | TimeGAN | **.107**±**.075** | **.160**±**.113** | .270±.106 | .320±.120 |
| | WaveGAN | .362±.080 | .345±.113 | .377±.099 | .385±.060 |
| | **TimeVAE** | **.118**±**.110** | **.140**±**.053** | **.167**±**.040** | **.068**±**.051** |
| | Diffusion-TS | .204±.086 | **.173**±**.063** | **.151**±**.055** | **.122**±**.051** |
| | Time-Transformer | .198±.169 | **.179**±**.116** | .408±.137 | .500±.000 |
| Sine | **RVAE-ST (ours)** | .069±.015 | **.113**±**.059** | **.080**±**.044** | **.021**±**.013** |
| | TimeGAN | .465±.130 | .457±.050 | .491±.005 | .497±.005 |
| | WaveGAN | .187±.036 | .367±.073 | .449±.025 | .449±.034 |
| | TimeVAE | .161±.092 | **.160**±**.124** | .272±.129 | .347±.144 |
| | Diffusion-TS | **.035**±**.014** | **.182**±**.163** | .294±.109 | .428±.105 |
| | Time-Transformer | .173±.019 | .491±.004 | .499±.001 | .500±.000 |
| MetroPT3 | RVAE-ST | **.098**±**.066** | .367±.109 | .423±.074 | .496±.004 |
| | TimeGAN | .428±.041 | .498±.002 | .499±.001 | .499±.001 |
| | WaveGAN | .432±.042 | .494±.005 | .497±.002 | .497±.003 |
| | TimeVAE | .279±.103 | .438±.070 | .488±.024 | .495±.004 |
| | **Diffusion-TS** | .139±.025 | **.251**±**.022** | **.319**±**.015** | **.486**±**.012** |
| | Time-Transformer | .473±.007 | .493±.005 | .500±.000 | .500±.000 |

dataset exhibits more nuanced behavior: Diffusion-TSperforms best at $l = 100$; at $l = 300$, RVAE-ST, TimeVAE, and Diffusion-TSperform comparably; and from $l = 500$ onwards, RVAE-ST achieves the best results. For the MetroPT3 dataset, RVAE-ST is best at $l = 100$, while Diffusion-TS slightly outperforms all other models at longer sequence lengths.

## 5.6 Evaluation by PCA and t-SNE

In this section, we evaluate the quality of the generated time series using dimensionality reduction techniques such as PCA (Hotelling, 1933) and t-SNE (Hinton & Van Der Maaten, 2008). The idea is to first train these methods on the original data, project the data into a lower-dimensional space, and visualize the resulting patterns. Subsequently, the same transformations are applied to the synthetic data to assess how well they align with the distribution of the original data. While these techniques are widely used and helpful for identifying structural similarities, it is important to note that they do not account for temporal dependencies within the sequences.

These common techniques complement earlier methods that primarily assessed the sample quality of the models. For brevity, we present the results of four selected experiments in the main paper, as all experiments consistently yield the same findings. These four experiments include PCA plots on the EM dataset and on the ECG dataset, each with sequence lengths of $l = 100$ and $l = 1000$. The full set of experiments is provided in Appendix A.8.

The visual inspection of the PCA plots for the EM dataset with a sequence length of $l = 100$ reveals no significant differences in the distributions of the models, with Time-Transformer showing a slightly less pronounced overlap compared to the other models. However, as the sequence length increases to l = 1000, the performance differences between the models become clearly visible. Interestingly, the PCA at this length exhibits a circular pattern, indicating the periodic characteristics of the dataset. Among the models, RVAE-ST demonstrates the highest degree of overlap between the original and synthetic data, fitting the circular pattern without outliers. Diffusion-TS performs almost equally well, with slightly less overlap compared to RVAE-ST (see figure 1 for visual comparison of the models). WaveGAN shows only a few outliers near the circular pattern. TimeVAE synthetic points further fill the circle, leading to greater deviation from the original data distribution.

The PCA plots for the ECG dataset provide a detailed view of models' performances. At l = 100, RVAE-ST, TimeVAE, and Diffusion-TS perform equally well, showing a strong overlap with the original data. WaveGAN and Time-Transformer show less overlap, and TimeGAN demonstrates almost no overlap at all. At l = 1000, RVAE-STachieves the best performance, with the original data being very well represented. This is followed by WaveGAN and TimeVAE, where the synthetic data points cluster together, but with less coverage of the original distribution. Diffusion-TS performs noticeably worse, while TimeGAN and Time-Transformer show almost no overlap, with the generated data exhibiting minimal variability.

## 5.7 Training scheme ablations

In this experiment, we compare the effectiveness of our proposed training approach against the conventional training method on the same network topology. Our comparison metric is the Evidence Lower Bound (ELBO), calculated for the original dataset $X \in \mathbb{R}^{n_s \times l \times c}$ where $n_s$ represents the numbers of samples, $l$ denotes the sequence length, and $c$ the number of channels. It is analogous to (4), but we get

$$\mathcal{E}(X) = \frac{1}{n_s} \sum_{i=0}^{n_s-1} \mathrm{ELBO}_{\mathrm{norm}} \left( \tilde{\mathcal{L}}_{\theta,\phi}(X_i) \right), \tag{5}$$

since we use the trained model $\tilde{\mathcal{L}}_{\theta,\phi}$ itself for evaluation. Simply speaking, it is the typical model evaluation on a dataset, but converted to $\mathrm{ELBO}_{\mathrm{norm}}$. We run this comparison on all datasets with a sequence length of 1000, which is particularly long and challenging. It is the maximum sequence length used in any of the previous experiments. For each of the following training schemes, we do 10 repetitions:

(i) **Conventional train**: One trains the model for a predefined sequence length of $l = 1000$

(ii) **Subsequent train**: The training procedure begins with a sequence length of $l = 100$ and continues until the stopping criteria are met. Afterward, we increase the sequence length by 100 and retrain the model, repeating this process until we complete training with a sequence length of $l = 1000$.

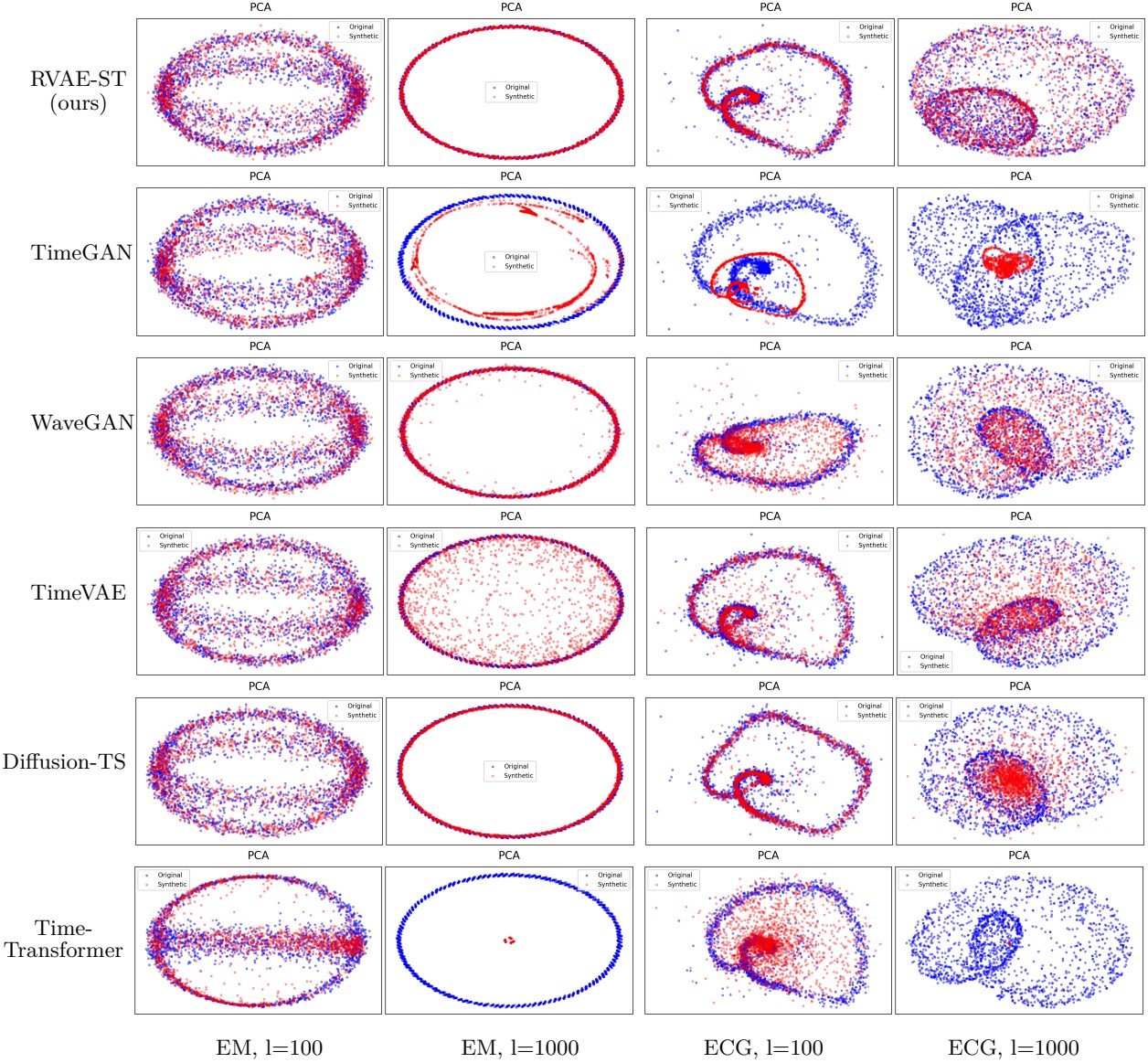

Figure 4: PCA plots for the EM and ECG datasets at sequence lengths of l = 100 and l = 1000. For the EM dataset, at l = 100, no significant differences are observed in the distributions of the models, with Time-Transformer showing a slightly less pronounced overlap. At l = 1000, the circular pattern of the data becomes more apparent, with RVAE-ST demonstrating the best performance, closely followed by Diffusion-TS. WaveGAN and TimeVAE show a few outliers and deviations, while TimeGAN exhibits almost no overlap. For the ECG dataset, at l = 100, RVAE-ST, TimeVAE, and Diffusion-TS show strong overlap with the original data, while WaveGAN and Time-Transformer exhibit less overlap, and TimeGAN shows almost no overlap. At l = 1000, RVAE-STperforms best, followed by WaveGAN and TimeVAE, with Diffusion-TS performing worse and TimeGAN and Time-Transformer showing minimal variability and no significant overlap.

As shown in Table 4, the subsequent training scheme (ii) consistently outperforms the conventional training scheme (i) across all datasets, with statistically significant improvements ($p < 0.002$). The largest performance gain is observed on the Sine dataset, where the model's ability to capture sinusoidal patterns improves substantially.

Table 4: Comparison of the effectiveness of our proposed training approach versus the conventional method. The performance metric is the $\text{ELBO}_{\text{norm}}$ as described in Appendix A.3. On each dataset and model we repeated the experiments $n = 10$ times. The 1-sigma confidence intervals describe the results between the independently trained models.

| Train method | EM | ECG | ETTm2 | Sine | MetroPT3 |
|---|---|---|---|---|---|
| conventional train | 0.094±0.004 | 0.103±0.000 | 0.174±0.016 | -0.837±0.566 | -0.140±0.061 |
| subsequent train | **0.218±0.004** | **0.201±0.004** | **0.217±0.012** | **0.194±0.010** | **0.142±0.019** |

## 6  Discussion

In this paper, we introduced a novel approach that utilizes a translation-equivariant network architecture to learn long sequence time series data. The key idea behind our approach is that stationary time series exhibit consistent patterns over time. Equivariance helps the model generalize better across time shifts, which is essential for capturing the underlying structure in stationary data. While LSTM layers contribute approximately to equivariance, the rest of the network's structure is fully equivariant. This approach is characterized by two key components: (1) an inductive bias tailored for time series with quasi-stationary behavior, and (2) the fact that the number of trainable parameters remains independent of the sequence length, enabling the model to effectively learn as the sequence length increases during training. These two key components enable the model to actively exploit the inherent stationarity of the data, leading to more efficient and scalable learning. In our experiments, we compared our model (RVAE-ST) with other state-of-the-art models across five different datasets. Three of these datasets exhibited stronger stationarity (Electric Motor, ECG, and Sine), while the remaining two (ETT and MetroPT3) displayed more dynamic behavior, though still retaining periodic patterns typical for sensor-based time series data. The results show that on the more stationary datasets, as the sequence length increases, our model significantly outperforms the other models, particularly in terms of FID and Discriminative Score. On the datasets with more dynamic behavior, our model also demonstrated competitive performance. The *average ELBO score* confirmed these findings, though it required a more nuanced interpretation due to its inherent limitations. The PCA and t-SNE plots further support our results . In Section A.1, we show that our model, with trained weights for a sequence length of $l = 1000$, can generate samples of arbitrary length on the three more stationary datasets. We have demonstrated this for a sequence length of $l = 5000$. This experiment strongly supports our initial intuition (see equation 2) that, for long time series, the hidden and cell states converge in a model with a translation-equivariant network topology. Our findings not only validate the effectiveness of our approach but also point to several promising directions for future work. The proposed methodology could potentially be extended to other model classes, such as diffusion-based generative models. Moreover, the stepwise increase in sequence length during training was chosen in a relatively ad-hoc manner; optimizing this training scheme could further enhance both performance and training efficiency. Finally, it would be valuable to investigate whether our approach can also be applied in different scaling regimes, for example by explicitly optimizing for shorter target sequence lengths, while still leveraging the benefits of the scalable architecture.

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

# A  Appendix

## A.1  Extended Time Series

In this section, we provide qualitative examples of generated time series for each of the five datasets used in our evaluation: Electric Motor, ECG, ETT, Sine, and MetroPT3. All samples were generated with a fixed sequence length of $l = 5000$, using model weights trained on sequences up to $l = 1000$. This allows us to assess the model's ability to generalize and synthesize plausible data beyond the training horizon.

The results illustrate how well the model maintains the structure of the original data when generating extended sequences:

- For datasets with stronger stationarity (Electric Motor, ECG, and Sine), the key patterns continue to be synthesized plausibly beyond the training length. In these cases, a stable state emerges, characterized by repeating, but not identical, patterns (see Figures 5, 6, and 8).

- In the Sine dataset, sinusoidal curves are extended effectively, with only a slight reduction in amplitude observable in some channels. (Figure 8).

- For the less stationary datasets (ETT and MetroPT3), a clear degradation in synthesis quality is observed beyond the trained length. In both cases, the model produces repetitive, flatline-like patterns with low variation, and characteristic structures are no longer preserved (Figures 7 and 9).

These qualitative results support the quantitative findings and further highlight the model's ability to generalize well on quasi-stationary data, while revealing its limitations on more dynamic datasets.

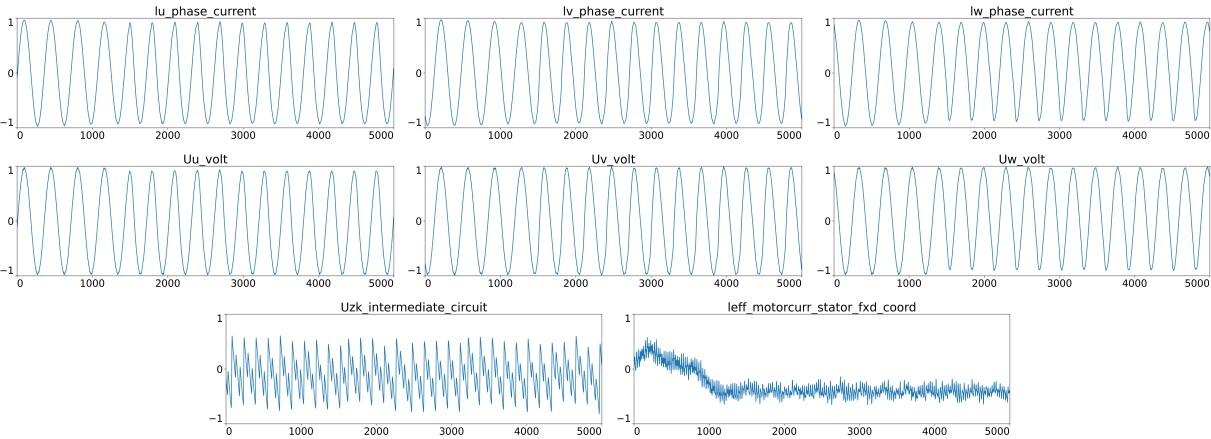

Figure 5: Example of a generated time series sample of length $l = 5000$ from the Electric Motor dataset. The model was trained on sequences up to $l = 1000$. The main characteristics of the dataset continue to be well synthesized in the extended sample. During generation, the model reaches a stable state in which the output patterns kind of repeat. As a result, slower trends, especially visible in the *leff motorcurr stator fxd coord* channel, are not fully reflected in the synthesis.

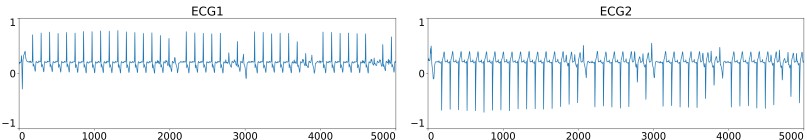

Figure 6: Example of a generated time series sample of length $l = 5000$ from the two-channel ECG dataset. The model was trained on sequences up to $l = 1000$. The key characteristics of the data, particularly the heartbeat-like patterns across both channels, continue to be well synthesized in the extended sequence. Still, a stable state emerges, with periodic patterns that, while not identical, remain strongly similar over time.

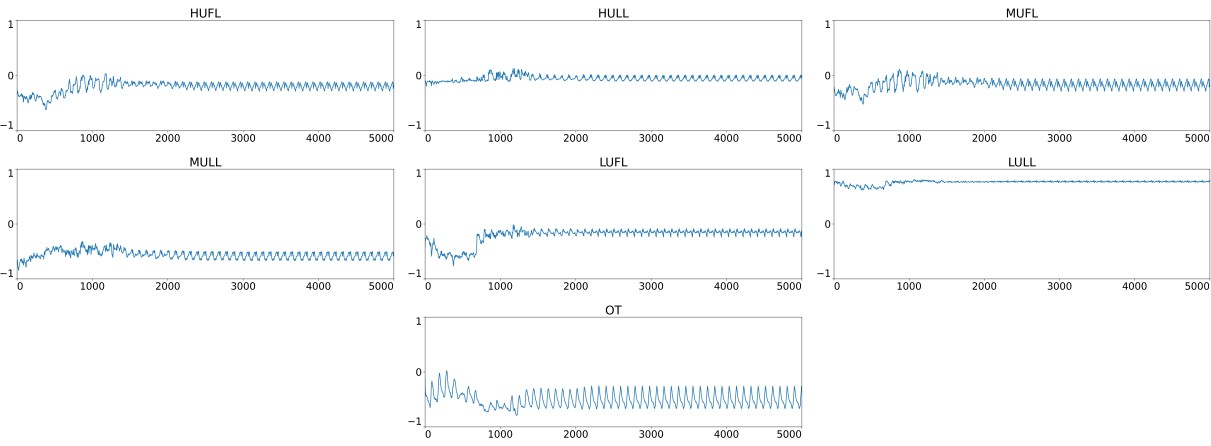

Figure 7: Example of a generated time series sample of length $l = 5000$ from the ETT dataset. The model was trained on sequences up to $l = 1000$. Up to this length, the synthesis closely follows the patterns present in the original data. Beyond this point, a stable state emerges. Most channels no longer reflect the dataset's characteristic patterns, though the "OT" channel still produces plausible structures.

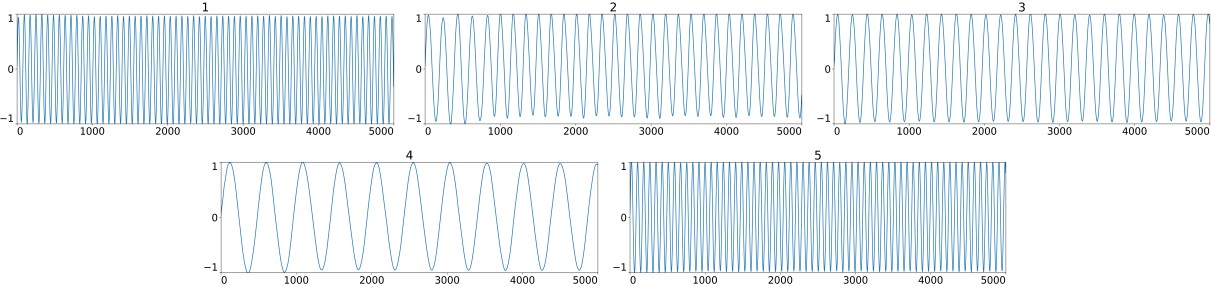

Figure 8: Example of a generated time series sample of length $l = 5000$ from the Sine dataset. The model was trained on sequences up to $l = 1000$. The sine curves are extended very consistently beyond the trained length, maintaining the dataset's structure. Upon closer inspection, a slight decrease in amplitude can be observed in channels 2 and 4 compared to the initial segment (up to $l = 1000$).

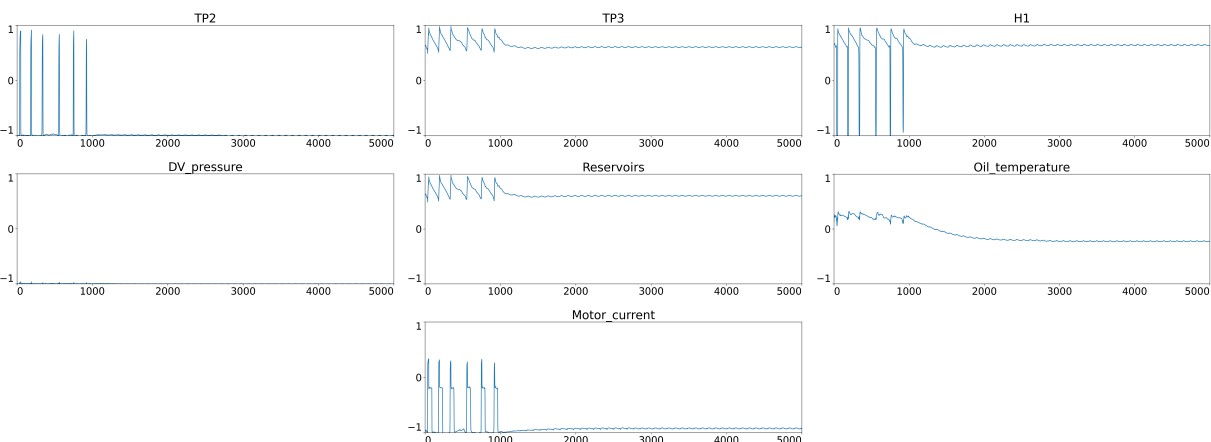

Figure 9: Example of a generated time series sample of length $l = 5000$ from the MetroPT3 dataset. The model was trained on sequences up to $l = 1000$. While the generation follows the original data up to this length, no meaningful structure is preserved in the extended part. Still, a stable state emerges, with the model settling into repetitive, low-variation patterns resembling noisy flatlines across all channels. This behavior is expected, as the MetroPT3 dataset exhibits low stationarity.

## A.2 Hyperparameters and Loss Function

In all experiments, for the encoder aswell as the decoder, we stack 4 LSTM-layers each with 256 hidden units. The latent dimension is $z = 20$. We use Adam optimizer with learning rate $\alpha = 10^{-4}$, $\beta_1 = 0.9$, $\beta_2 = 0.999$, $\epsilon = 10^{-7}$. We perform min-max scaling with $(-1, 1)$. After scaling we do a train/validation split with a ratio of 9:1.

We use the loss function

$$\mathcal{L}_{\theta,\phi} = \alpha \cdot \text{SSE} + \beta \cdot \text{D}_{\text{KL}}, \tag{6}$$

where the reconstruction loss, SSE, represents the sum of squared errors, computed for each individual sample within a batch:

$$\text{SSE} = \sum_T \sum_C (y_{tc} - \hat{y}_{tc})^2, \tag{7}$$

where $T$ is the sequence length and $C$ is the number of channels. We then average the SSE over the entire batch. In our experiments we set $\alpha = \frac{500}{T}$ and $\beta = 0.1$.

The parameter $\beta$ was introduced with the $\beta$-VAE (Higgins et al., 2017). For $0 < \beta < 1$ the VAE stores more bits about each input and the reconstructed sample is less smoothed out. If $\beta > 1$ the VAE is encouraged to learn a latent representation that is disentangled (Burgess et al., 2018). We adjust $\alpha$ antiproportional to the sequence length to retain the ratio between the reconstruction loss and the KL-Divergence.

### A.3 Loss to ELBO conversion

Transforming the VAE loss function into the Evidence Lower Bound (ELBO) is essential to connect the optimization process to a well-established probabilistic framework. The ELBO arises from the variational inference approach, which allows us to approximate the intractable posterior distribution of latent variables by optimizing a lower bound to the marginal likelihood of the observed data. By expressing the VAE loss as the ELBO, we clarify that the model's objective is twofold: maximizing the likelihood of the data through reconstruction and simultaneously regularizing the latent space by minimizing the divergence between the approximate posterior and the prior distribution. This dual objective ensures that the learned latent space reflects meaningful, structured representations while maintaining the ability to reconstruct the input data. Using the ELBO as the loss function thus ties the VAE training to a coherent probabilistic theory, enhancing both its interpretability and its ability to generate diverse and realistic data.

Given the likelihood,

$$\text{likelihood} = \prod_T \prod_C \frac{1}{\sqrt{2\pi\sigma^2}} \exp\left(-\frac{1}{2}\frac{(y_{tc} - \hat{y}_{tc})^2}{\sigma^2}\right), \tag{8}$$

we compute the log-likelihood, which can then be reformulated in terms of the SSE:

$$
\begin{aligned}
\text{log-likelihood} &= \log\left(\prod_T \prod_C \frac{1}{\sqrt{2\pi\sigma^2}} \exp\left(-\frac{1}{2}\frac{(y_{tc} - \hat{y}_{tc})^2}{\sigma^2}\right)\right) \\
&= \sum_T \sum_C \log\left(\frac{1}{\sqrt{2\pi\sigma^2}} \exp\left(-\frac{1}{2}\frac{(y_{tc} - \hat{y}_{tc})^2}{\sigma^2}\right)\right) \\
&= \sum_T \sum_C \left(\log\left(\frac{1}{\sqrt{2\pi\sigma^2}}\right) + \log\left(\exp\left(-\frac{1}{2}\frac{(y_{tc} - \hat{y}_{tc})^2}{\sigma^2}\right)\right)\right) \\
&= \sum_T \sum_C \left(\log\left(\frac{1}{\sqrt{2\pi\sigma^2}}\right) - \frac{1}{2}\frac{(y_{tc} - \hat{y}_{tc})^2}{\sigma^2}\right) \\
&= \sum_T \sum_C \left(-\frac{1}{2}\log(2\pi\sigma^2) - \frac{1}{2}\frac{(y_{tc} - \hat{y}_{tc})^2}{\sigma^2}\right) \\
&= -\frac{1}{2}\log\left(2\pi\sigma^2\right) \cdot T \cdot C - \frac{1}{2\sigma^2}\sum_T \sum_C (y_{tc} - \hat{y}_{tc})^2 \\
&= -\frac{1}{2}\log\left(2\pi\sigma^2\right) \cdot T \cdot C - \frac{1}{2\sigma^2}\text{SSE} \\
\Longleftrightarrow -\frac{1}{2\sigma^2}\text{SSE} &= \text{log-likelihood} + \frac{1}{2}\log\left(2\pi\sigma^2\right) \cdot T \cdot C \\
\Longleftrightarrow \text{SSE} &= -2\sigma^2 \cdot \text{log-likelihood} - \sigma^2 \log\left(2\pi\sigma^2\right) \cdot T \cdot C.
\end{aligned}
\tag{9}
$$

The ELBO is defined as the log-likelihood minus the kl-divergence (Murphy, 2022):

$$\text{ELBO} = \text{log-likelihood} - D_{\text{KL}}. \tag{10}$$

Given (6), (9) and $\sigma^2 = 0.5 \cdot \frac{\beta}{\alpha}$, we can derive the conversion to the ELBO:

$$
\begin{aligned}
\frac{\mathcal{L}_{\theta,\phi}}{\beta} &= \frac{\alpha}{\beta} \cdot \text{SSE} + \text{D}_{\text{KL}} \\
&= \frac{\alpha}{\beta} \left( -2\sigma^2 \cdot \text{log-likelihood} - \sigma^2 \cdot \log\left(2\pi\sigma^2\right) \cdot T \cdot C \right) + \text{D}_{\text{KL}} \\
&= -2\sigma^2 \cdot \frac{\alpha}{\beta} \cdot \text{log-likelihood} - \sigma^2 \cdot \frac{\alpha}{\beta} \cdot \log\left(2\pi\sigma^2\right) \cdot T \cdot C + \text{D}_{\text{KL}} \\
&= -2 \cdot 0.5 \cdot \frac{\beta}{\alpha} \cdot \frac{\alpha}{\beta} \cdot \text{log-likelihood} - 0.5 \cdot \frac{\beta}{\alpha} \cdot \frac{\alpha}{\beta} \cdot \log\left(2\pi \cdot 0.5 \cdot \frac{\beta}{\alpha}\right) \cdot T \cdot C + \text{D}_{\text{KL}} \\
&= -\text{log-likelihood} - 0.5 \cdot \log\left(\pi \cdot \frac{\beta}{\alpha}\right) \cdot T \cdot C + \text{D}_{\text{KL}} \\
\iff \text{log-likelihood} - \text{D}_{\text{KL}} &= -\frac{\mathcal{L}_{\theta,\phi}}{\beta} - 0.5 \cdot \log\left(\pi \cdot \frac{\beta}{\alpha}\right) \cdot T \cdot C \\
\implies \text{ELBO}(\mathcal{L}_{\theta,\phi}, \alpha, \beta, T, C) &= -\frac{\mathcal{L}_{\theta,\phi}}{\beta} - 0.5 \cdot \log\left(\pi \cdot \frac{\beta}{\alpha}\right) \cdot T \cdot C.
\end{aligned}
\tag{11}
$$

In our experiments, we normalize the ELBO by dividing it by the product of the number of channels and the sequence length. This normalization allows for a fairer comparison of model performance across datasets with different dimensionalities, such as varying sequence lengths or numbers of channels. Without this adjustment, the ELBO would scale with the size of the data, potentially biasing the evaluation in favor of datasets with larger sequences or more channels. By normalizing, we make the ELBO more independent of the specific data structure, enabling a more consistent comparison of the underlying model's ability to capture data patterns.

Although this normalization provides a useful heuristic for comparing different datasets, it should be noted that it does not guarantee perfect comparability in all cases. In some situations, larger datasets with more channels or longer sequences may introduce additional complexity, which could influence the model's performance. Therefore, while the normalized ELBO serves as a practical and interpretable metric.

We denote the normalized version of the ELBO as

$$
\text{ELBO}_{\text{norm}}(\mathcal{L}_{\theta,\phi}, \alpha, \beta, T, C) = \frac{\text{ELBO}(\mathcal{L}_{\theta,\phi}, \alpha, \beta, T, C)}{T \cdot C}.
\tag{12}
$$

### A.4 Implementation details of comparison models

### A.4.1 Global hyperparameters

To balance data diversity and computational efficiency, we used a dataset-specific step size when splitting time series into training sequences. This step size determines the offset between starting points of consecutive sequences, thereby influencing both the number of training samples and the memory requirements during training.

For the Electric Motor, ECG, and MetroPT3 datasets, we chose a step size of $0.1 \cdot l$, where $l$ is the sequence length. For the ETT dataset, which exhibits more complex and longer-range temporal dependencies, we used a smaller step size of $0.04 \cdot l$ to increase the number of training samples. In contrast, for the synthetic Sine dataset, we fixed the number of training samples to 10,000 for each sequence length.

This approach reflects a practical trade-off: while smaller step sizes increase training data diversity, they also lead to higher memory usage. Particularly for long sequences, using very small step sizes (e.g., step size = 1) can cause GPU memory overflow or even exceed system RAM, depending on the model architecture, implementation and dataset.

### A.4.2 TimeGAN

We did all experiments with the same hyperparameters. Num layers=3, hidden dim=100, num iterations = 25000. The clockwise computation time on these hyperparameters were the highest of all models. We use the authors original implementation[4] on a Nvidia DGX server in the 19.12-tf1-py3 container[5]. On sequence length $l = 1000$, the training took about 3 weeks wall-clock time.

### A.4.3 WaveGAN

For WaveGan needed special preperation to be usable for training. First we min maxed scaled the dataset file, split it into training and validation parts and then converted each into a n-dimensional *.wav* file. WaveGan is limited in configurability. In terms of sequence length the user can decide between $2^{14}$, $2^{15}$ and $2^{16}$. We chose $2^{14} = 16384$ because it is the smallest possible length. When we generate samples, we cut them into equal parts which correspond to the desired sequence length $l$. The rest of the hyperparameters were set to default. On the sine dataset training, wie used created 10,000 samples with a length of 16,384. We used the ported pytorch implementation[6].

### A.4.4 TimeVAE

We use TimeVAE with default parameters. We integrated components of the original TimeVAE implementation[7], such as the encoder, decoder, and loss function, into our own program framework. The reconstruction loss of TimeVAE is

$$\sum_T \sum_C (y_{tc} - \hat{y}_{tc})^2 + \frac{1}{C} \sum_C (\bar{y}_c - \bar{\hat{y}}_c)^2. \tag{13}$$

TimeVAEincludes a hyperparameter a, which acts as a weighting factor for the reconstruction loss. The authors of the original paper recommend using a value for a in the range of 0.5 to 3.5 to balance the trade-off between reconstruction accuracy and latent space regularization. In all of our experiments, we set $a = 3$.

### A.4.5 RCGAN

We used the original implementation[8] on a Nvidia DGX server in the 19.12-tf1-py3 container[9]. We trained RCGANfor 500 epochs and afterswards used the weights with the lowest $\hat{t}$ for sampling the dataset. On the ETT dataset, training was numerically instable for $l \geq 300$.

## A.5 Discriminative Score

The 2-layer RNN for binary classification consists of a GRU layer, where the hidden dimension is set to $\lfloor n_c/2 \rfloor$, where $n_c$ is the number of channels. This is followed by a linear layer with an output dimension of one. To prevent overfitting, early stopping with a patience of 50 is applied. We each discriminative score we repeated 15 training procedures. On each procedure, 2000 random samples were used as the train dataset and 500 samples were used as the validation dataset for early stopping monitoring. The discriminative score is then determined by validating further independent 500 samples.

## A.6 Average Elbo with TimeVAE Elbo-Model

Table 5 shows the results for the average *ELBO* score $\mathcal{E}(\tilde{X})$ using the base of TimeVAE as the *ELBO model*. However, instead of using the original loss function of TimeVAE, we utilized the loss function of RVAE-ST

---

[4]https://github.com/jsyoon0823/TimeGAN
[5]https://docs.nvidia.com/deeplearning/frameworks/tensorflow-release-notes/rel_19.12.html
[6]https://github.com/mostafaelaraby/wavegan-pytorch
[7]https://github.com/abudesai/timeVAE
[8]https://github.com/ratschlab/RGAN
[9]https://docs.nvidia.com/deeplearning/frameworks/tensorflow-release-notes/rel_19.12.html

as it simplifies the conversion to the *ELBO* score as shown in (11). Analogous to Table 2, our model is outperforming all other models. The Wilcoxon rank test indicates statistical significance with p <0.0001, except for the ETT dataset at $l = 100$ and $l = 300$, where TimeVAEis statistically better.

Table 5: **Average *ELBO* score** $\mathcal{E}(\tilde{X})$ of synthetic time series for six models (see 5.2), computed on the five datasets (see 5.1) at sequence lengths of $l = 100$, $l = 300$, $l = 500$, and $l = 1000$. A higher score indicates better performance. For each score, 1500 generated samples were evaluated by an *ELBO model* (based on the TimeVAE architecture) and the results are reported with 1-sigma confidence intervals. The interpretation must follow analogously to the explanation provided in Section 5.4 of the main paper, where the specifics and limitations of the ELBO score are discussed in detail.

| Dataset | Model | Sequence lengths | | | |
| | | 100 | 300 | 500 | 1000 |
|---------|-------|------|------|------|------|
| Electric Motor | **RVAE-ST (ours)** | **1.61±0.69** | **1.64±0.12** | **1.64±0.01** | **1.64±0.02** |
| | TimeGAN | 1.29±0.39 | 1.33±0.17 | 1.21±0.10 | -2.14±0.82 |
| | WaveGAN | 1.52±0.14 | 1.47±1.05 | 1.52±0.22 | 1.52±0.15 |
| | TimeVAE | 1.52±0.87 | 1.44±1.28 | 1.01±2.35 | 0.10±3.58 |
| | Diffusion-TS | 1.56±0.45 | 1.35±0.36 | 1.39±0.21 | 1.30±0.29 |
| | Time-Transformer | 1.25±1.88 | -22.9±7.52 | -85.4±18161 | -22.7±8.05 |
| ECG | **RVAE-ST (ours)** | 1.62±0.07 | 1.62±0.07 | 1.62±0.06 | 1.59±0.06 |
| | TimeGAN | -2.57±0.22 | -2.26±0.22 | -2.67±1.92 | -2.58±0.49 |
| | WaveGAN | 1.32±0.29 | 1.33±0.18 | 1.32±0.16 | 1.32±0.15 |
| | TimeVAE | 1.57±0.15 | 1.46±0.16 | 1.39±0.15 | 1.08±0.28 |
| | Diffusion-TS | **1.63±0.06** | 1.63±0.08 | 1.60±0.18 | 1.16±25.2 |
| | Time-Transformer | 1.22±0.50 | **1.67±0.04** | **1.67±0.04** | **1.67±0.04** |
| ETT | **RVAE-ST (ours)** | **1.56±0.24** | **1.57±0.09** | **1.59±0.05** | **1.60±0.13** |
| | TimeGAN | 1.49±0.17 | 1.20±1.49 | 0.83±0.91 | -0.00±0.28 |
| | WaveGAN | 1.50±0.50 | 1.50±0.41 | 1.47±0.64 | 1.49±0.43 |
| | TimeVAE | **1.56±0.45** | 1.41±0.81 | 1.15±1.05 | 0.40±2.06 |
| | Diffusion-TS | 1.53±0.07 | 1.52±0.13 | 1.52±0.13 | 1.52±0.16 |
| | Time-Transformer | 1.43±0.52 | **1.57±0.11** | 1.48±0.04 | -39.6±5.63 |
| Sine | RVAE-ST (ours) | **-34.5±29.1** | -48.7±27.8 | -51.9±27.5 | -56.2±28.3 |
| | TimeGAN | -40.1±32.6 | -54.8±31.2 | -61.6±28.6 | -62.9±46.5 |
| | WaveGAN | -41.6±27.9 | -49.0±26.7 | -51.2±27.4 | -51.9±27.1 |
| | TimeVAE | **-34.4±27.9** | -51.9±31.5 | -54.4±308 | -62.3±138 |
| | Diffusion-TS | -36.4±31.2 | -40.3±24.2 | -48.2±26.7 | -44.5±25.4 |
| | **Time-Transformer** | -38.1±30.1 | **-32.0±19.5** | **-2.60±5.32** | **-1.43±3.95** |
| MetroPT3 | RVAE-ST (ours) | 1.49±0.64 | 1.38±0.77 | 1.39±0.74 | **1.36±0.81** |
| | TimeGAN | 1.33±0.77 | 0.95±1.83 | 1.42±0.84 | -0.07±2.94 |
| | WaveGAN | 0.35±1.57 | 0.18±1.67 | 0.23±1.64 | 0.22±1.64 |
| | TimeVAE | 1.06±1.14 | -0.07±2.07 | -2.81±3.43 | -5.61±3.36 |
| | Diffusion-TS | **1.63±0.24** | **1.58±0.49** | **1.59±0.41** | 1.04±2.08 |
| | Time-Transformer | 0.06±1.73 | -0.97±2.21 | -1.29±0.63 | -331±26.2 |

## A.7  PyTorch vs TensorFlow

The experiments were conducted using a TensorFlow implementation of our model. Additionally, we performed tests with a PyTorch reimplementation (which is not part of this paper). In these tests, we found that the performance in PyTorch was significantly worse compared to the TensorFlow implementation.

Upon investigation, we identified the cause of the performance difference. The weight initialization in both the LSTM and Dense layers differs between TensorFlow and PyTorch. Specifically, TensorFlow uses a uniform distribution for the initialization of both LSTM and Dense weights, while PyTorch employs different initialization methods by default. To align the behavior between both frameworks, we modified the PyTorch implementation to use the same uniform weight initialization for both LSTM and Dense layers as in TensorFlow. After making these adjustments, we were able to achieve consistent results across both frameworks.

### A.8   PCA and t-SNE Results

The following section presents the PCA and t-SNE plots for all experiments, including each dataset, model, and sequence length. Since RCGAN consistently underperforms, and the performance of TimeGAN and WaveGAN remains unchanged across sequence lengths within a given dataset, these points will not be explicitly mentioned in each figure to maintain clarity and readability.

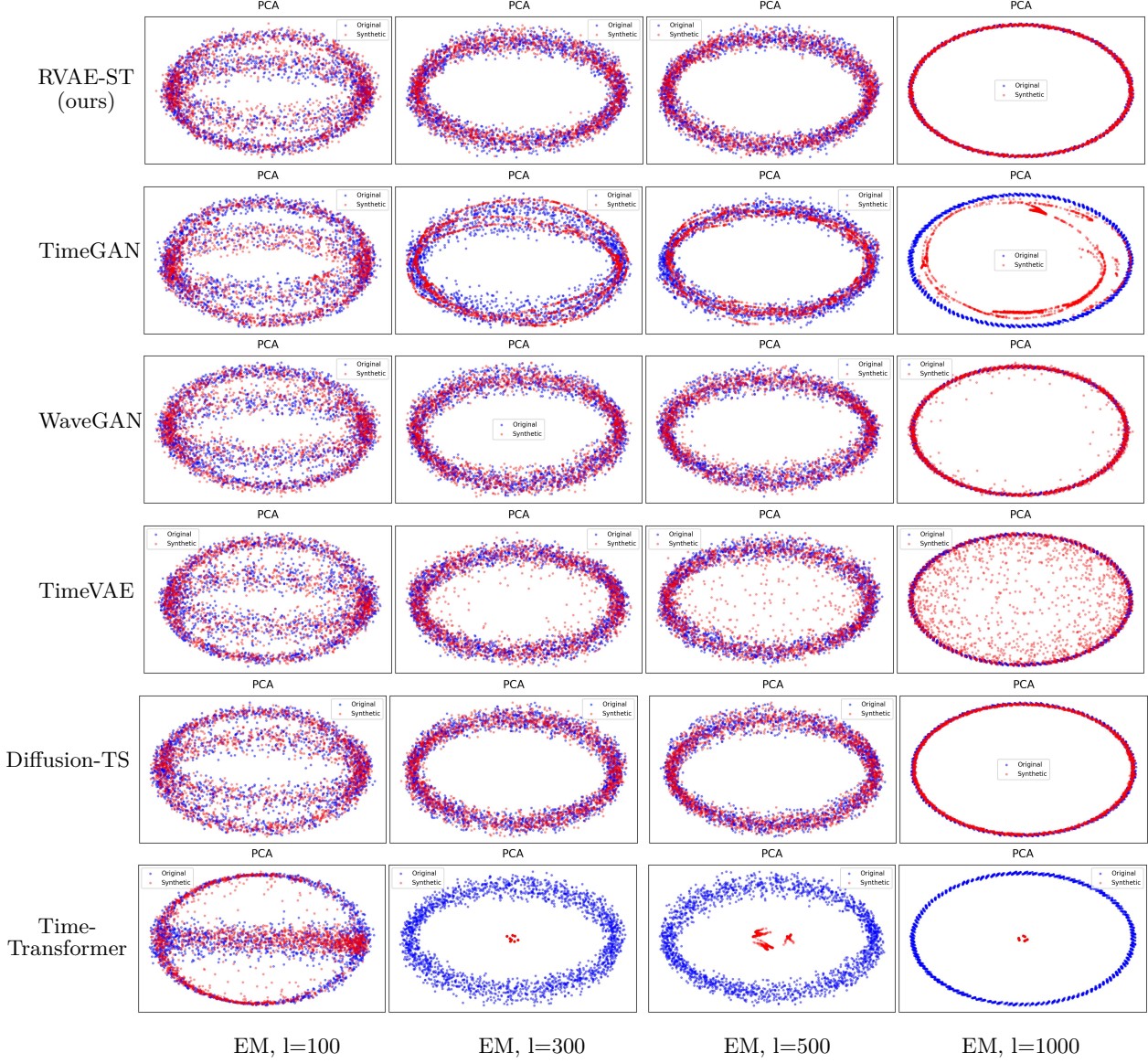

Figure 10: PCA plots for all sequence lengths on the Electric Motor dataset. At $l = 100$, all models perform similarly, though Time-Transformer already shows slightly weaker results. From $l = 300$ onward, TimeGAN and TimeVAE both degrade consistently with increasing sequence length, with TimeGAN showing reduced variance. Time-Transformer fails to generate coherent samples beyond this point. At $l = 1000$, RVAE-ST and Diffusion-TS produce the most consistent results, followed by WaveGAN.

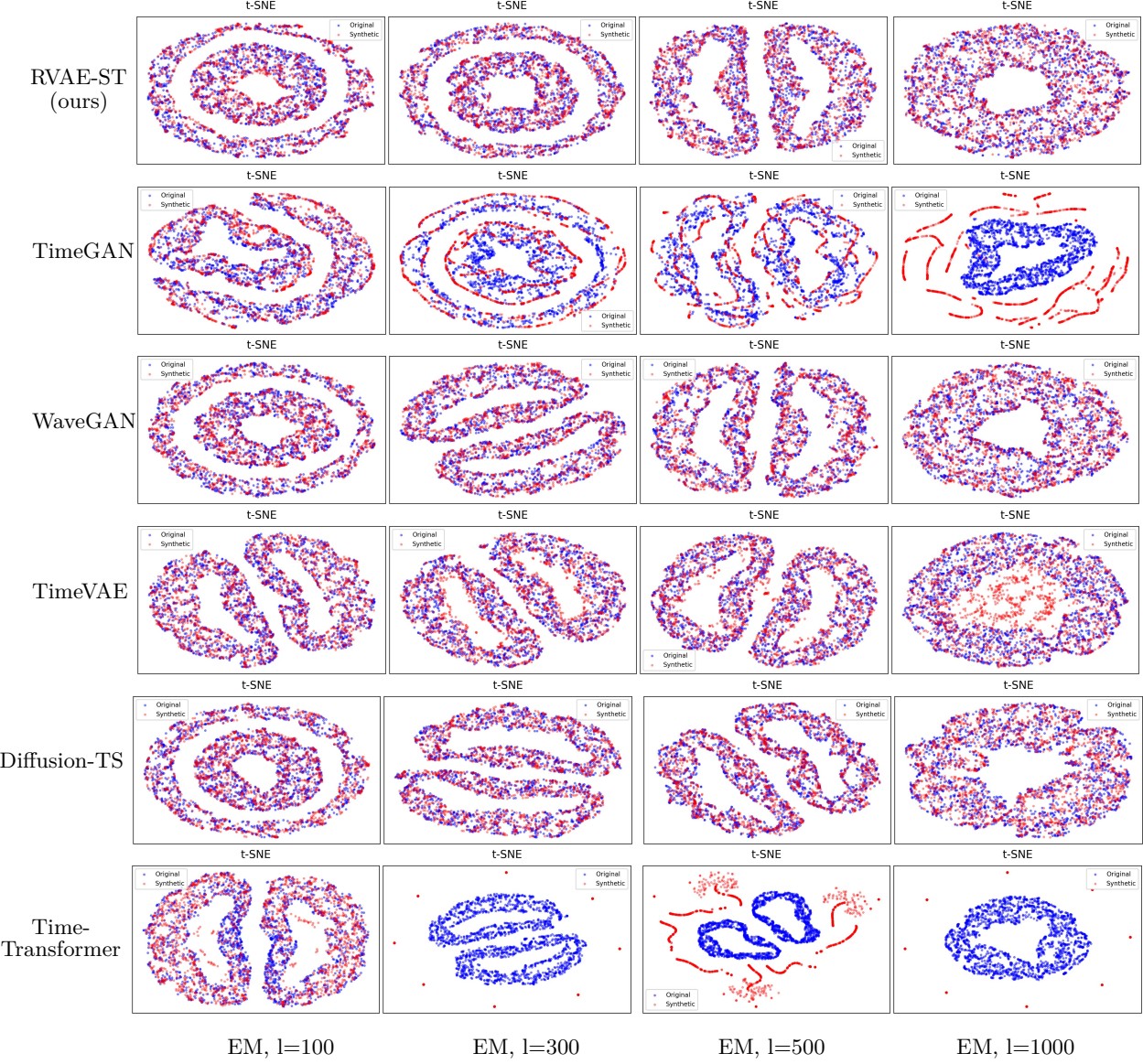

Figure 11: t-SNE plots for all sequence lengths on the Electric Motor dataset. At $l = 100$, TimeGAN already performs worse than the other models, similarly to Time-Transformer. From $l = 300$ onward, TimeGAN shows further deterioration, while TimeVAE also degrades but to a lesser extent. Time-Transformer fails to generate coherent samples at longer sequence lengths. At all sequence lengths, WaveGAN, RVAE-ST, and Diffusion-TS perform similarly.

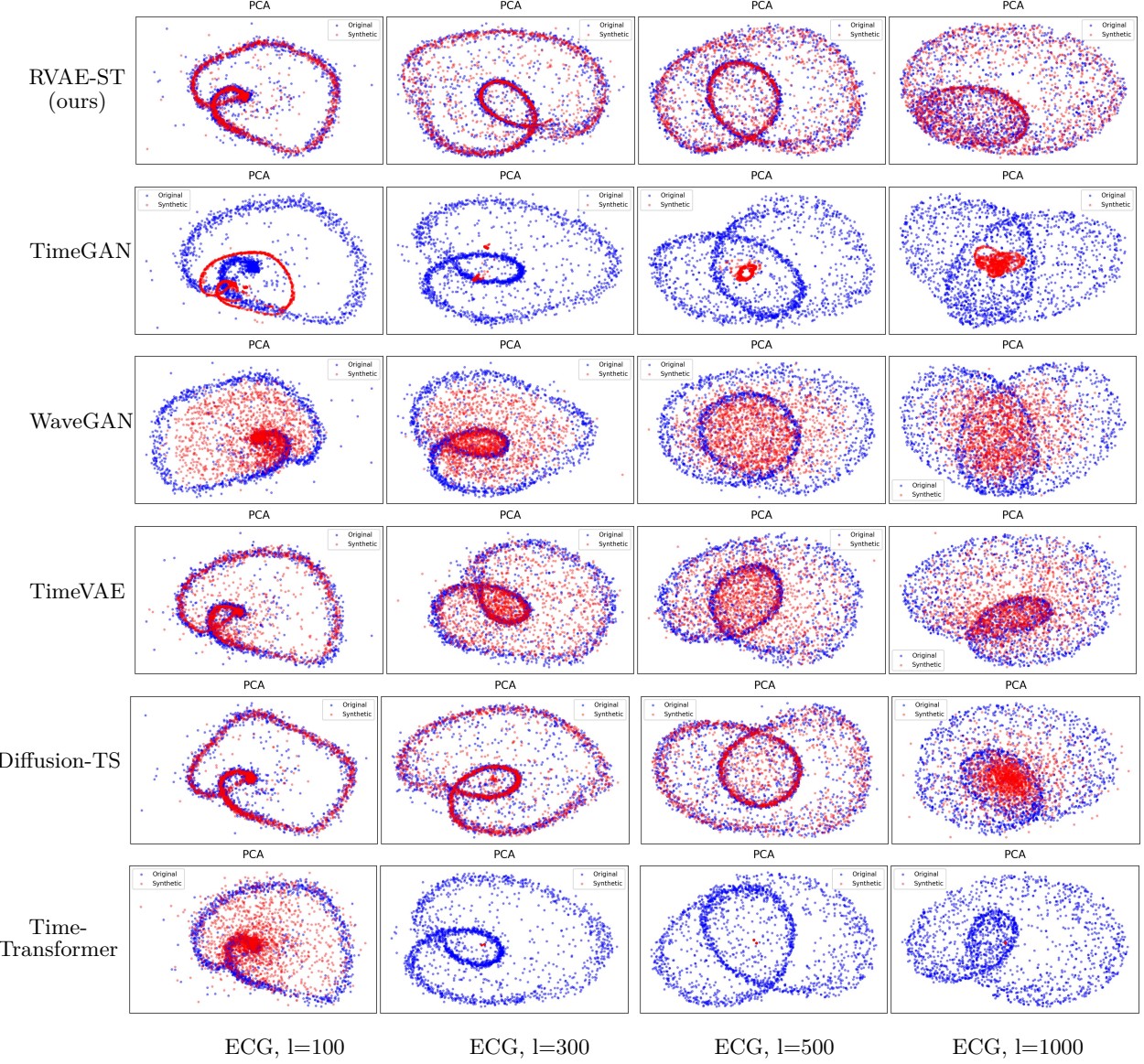

ECG, l=100          ECG, l=300          ECG, l=500          ECG, l=1000

Figure 12: PCA plots for all sequence lengths on the ECG dataset. At $l = 100$, TimeVAE performs similarly to RVAE-ST and Diffusion-TS. RVAE-ST shows the best performance at $l = 1000$. Diffusion-TS performs as well as RVAE-ST up to $l = 500$. WaveGAN consistently performs worse than the best models but still significantly outperforms TimeGAN and Time-Transformer , which fail to generate coherent samples starting from $l = 300$.

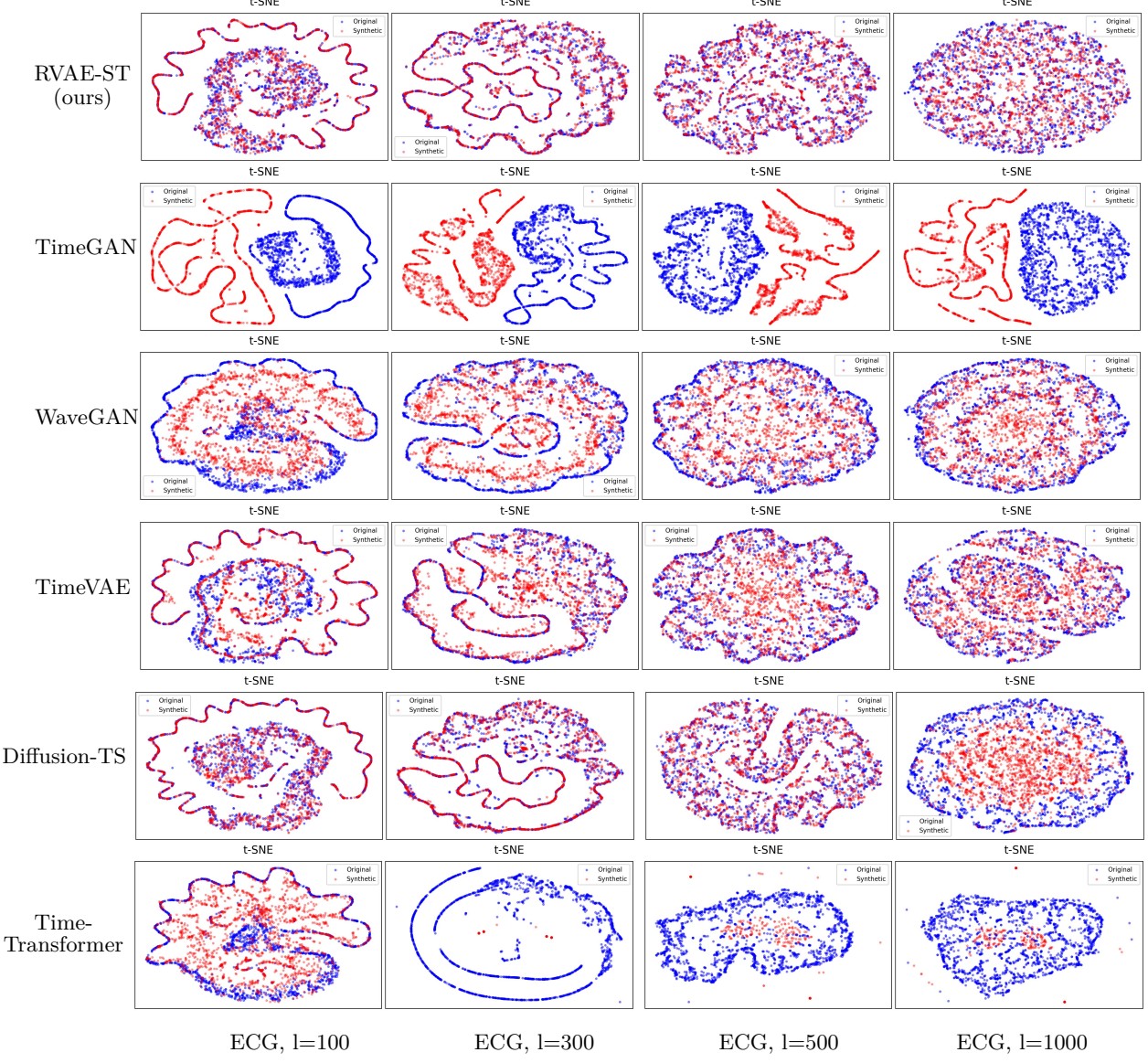

ECG, l=100          ECG, l=300          ECG, l=500          ECG, l=1000

Figure 13: t-SNE plots for all sequence lengths on the ECG dataset. At $l = 100$, TimeVAE performs similarly to RVAE-ST and Diffusion-TS. RVAE-ST shows the best performance at $l = 1000$. Diffusion-TS performs as well as RVAE-ST up to $l = 500$. WaveGAN consistently performs worse than the best models but still outperforms TimeGAN and Time-Transformer , which fail to generate coherent samples starting from $l = 300$.

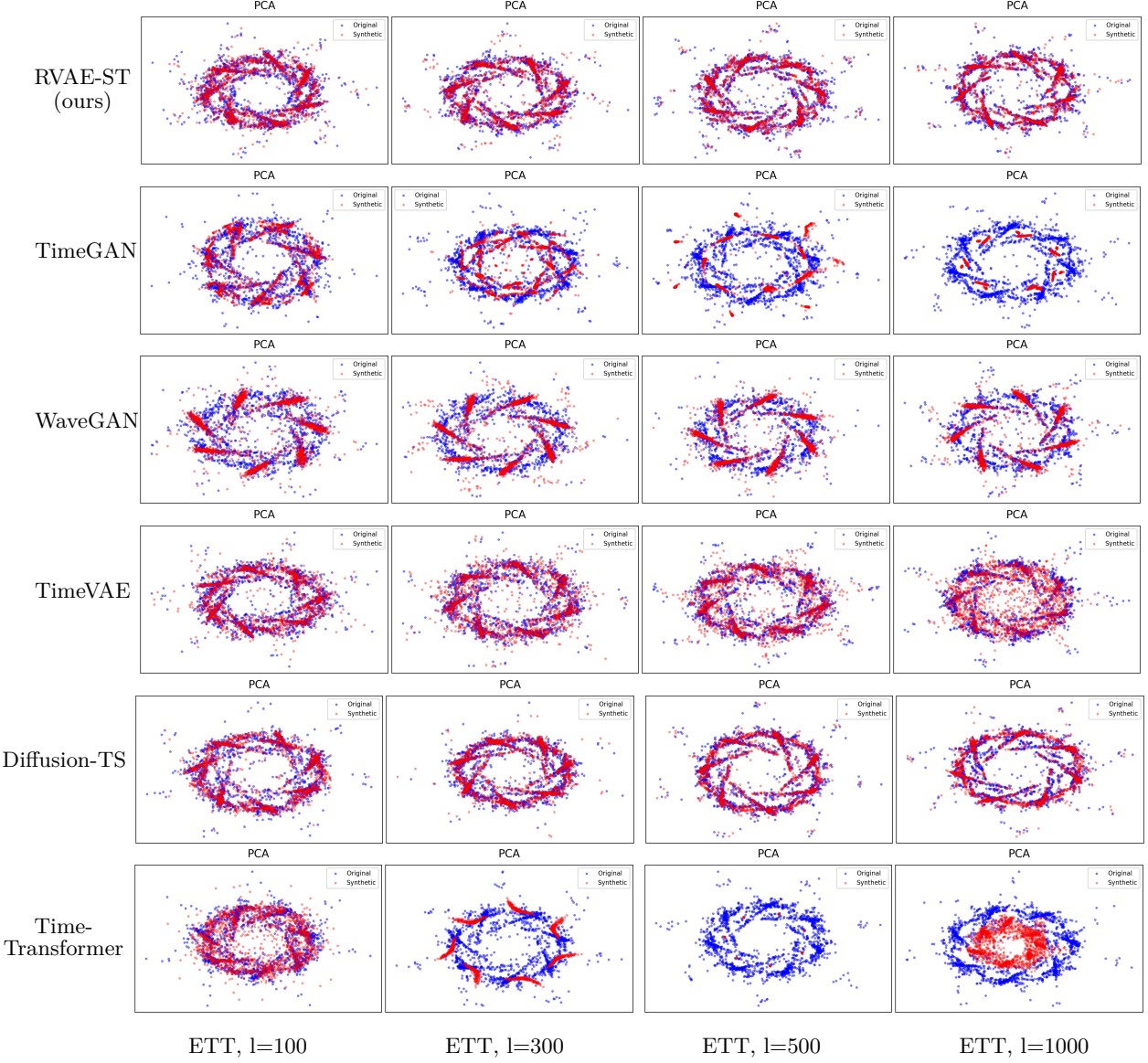

Figure 14: PCA plots for all sequence lengths on the ETT dataset. RVAE-ST and Diffusion-TS consistently perform the best across all sequence lengths. WaveGAN fails to capture the full variance of the dataset. TimeVAE performs similarly to RVAE-ST and Diffusion-TS at $l = 100$, but its performance degrades with increasing sequence length. TimeGAN and Time-Transformer perform reasonably well at $l = 100$, though already worse than the other models, and their performance significantly drops starting from $l = 300$.

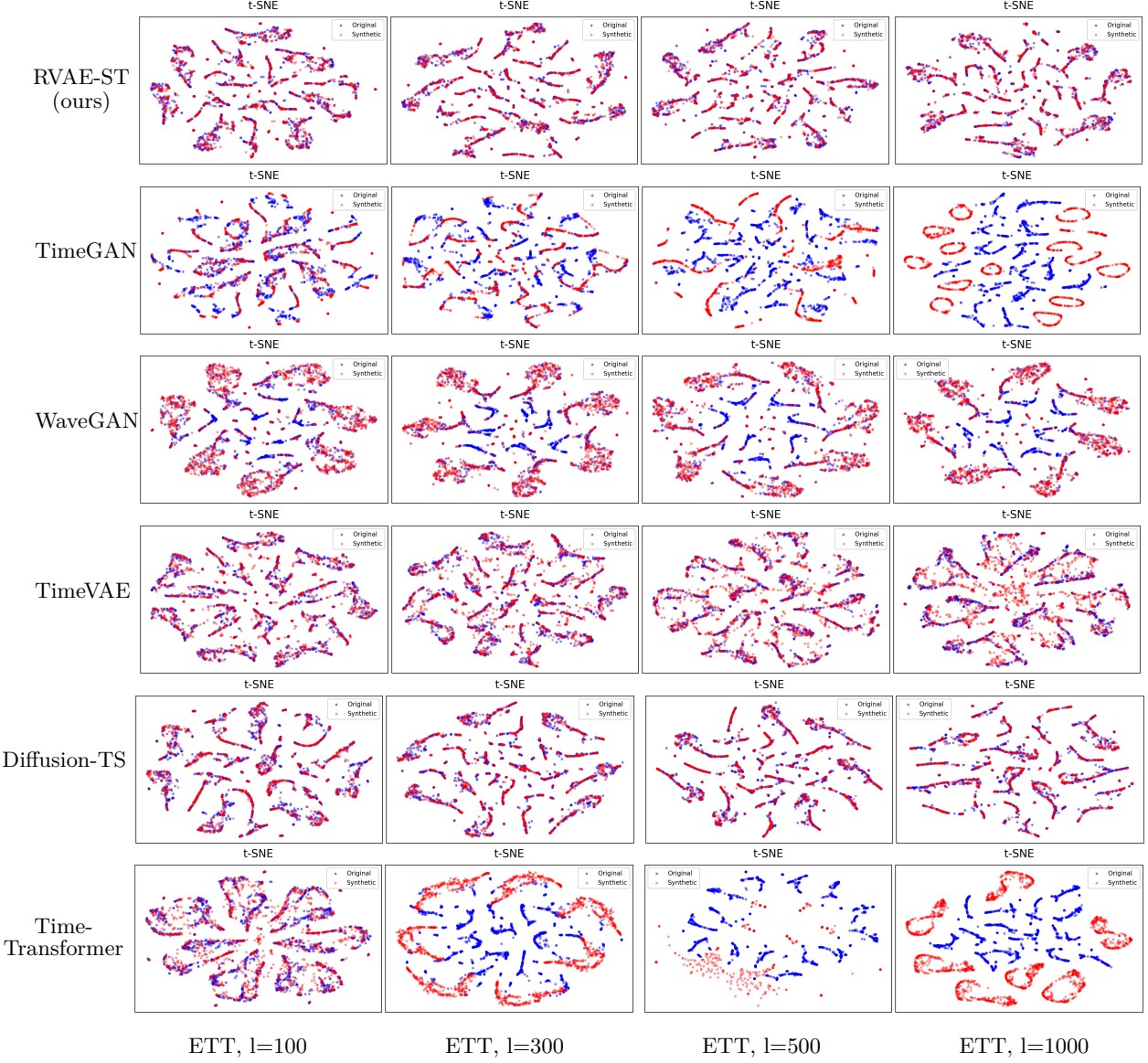

Figure 15: t-SNE plots for all sequence lengths on the ETT dataset. RVAE-ST and Diffusion-TS consistently perform the best across all sequence lengths. WaveGAN fails to capture the full variance of the dataset. TimeVAE performs similarly to RVAE-ST and Diffusion-TS at $l = 100$, but its performance degrades with increasing sequence length. TimeGAN and Time-Transformer perform reasonably well at $l = 100$, though already worse than the other models, and their performance significantly drops starting from $l = 300$.

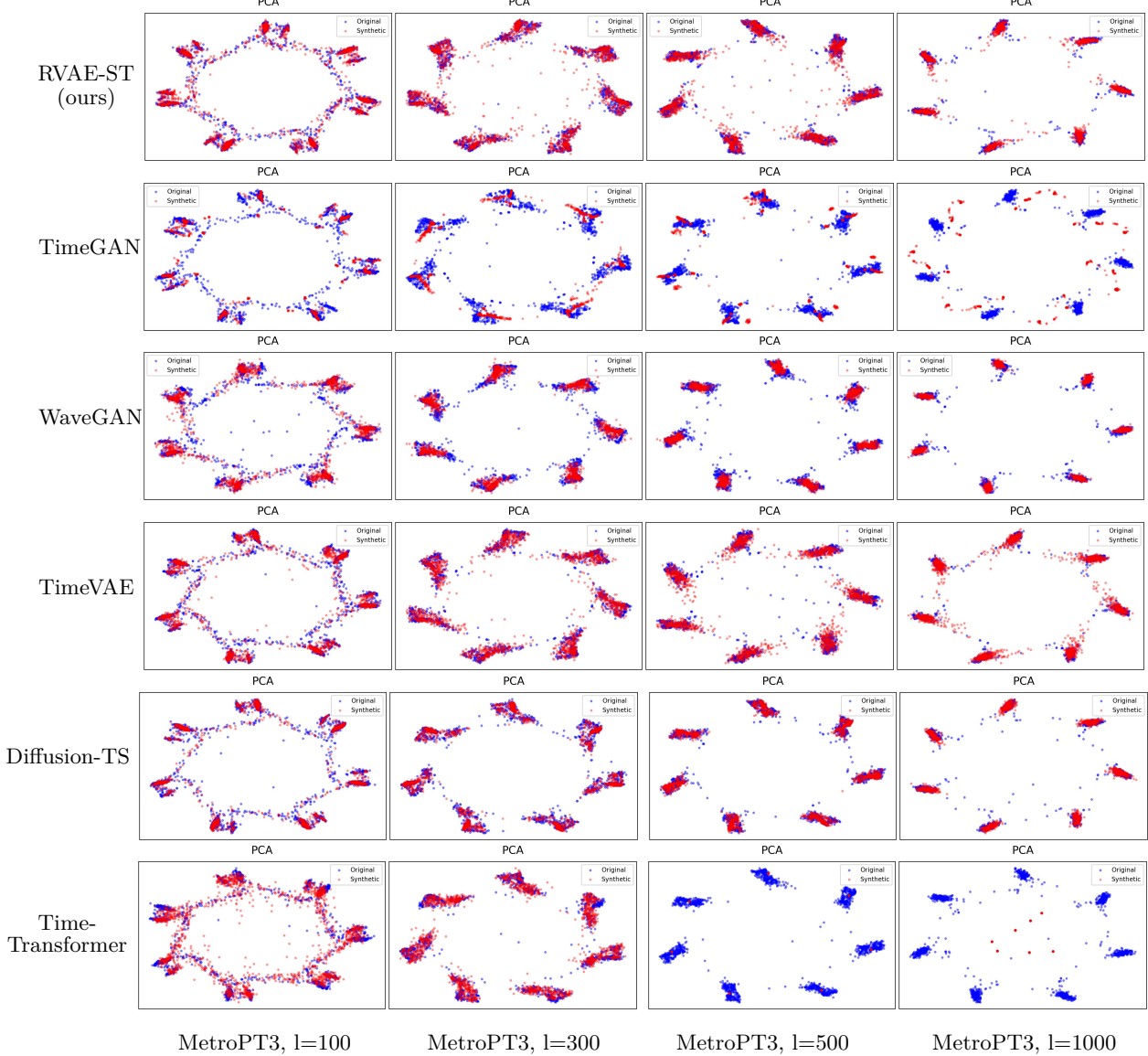

Figure 16: PCA plots for all sequence lengths on the MetroPT3 dataset. RVAE-ST, TimeVAE and Diffusion-TS perform similarly and the best across all sequence lengths. WaveGAN performs slightly worse, as it does not capture the entire distribution of the dataset (with a minimal difference). Time-Transformer performs reasonably well at $l = 100$ and $l = 300$, but its performance degrades at longer sequence lengths. TimeGAN consistently performs the worst, failing to generate plausible samples.

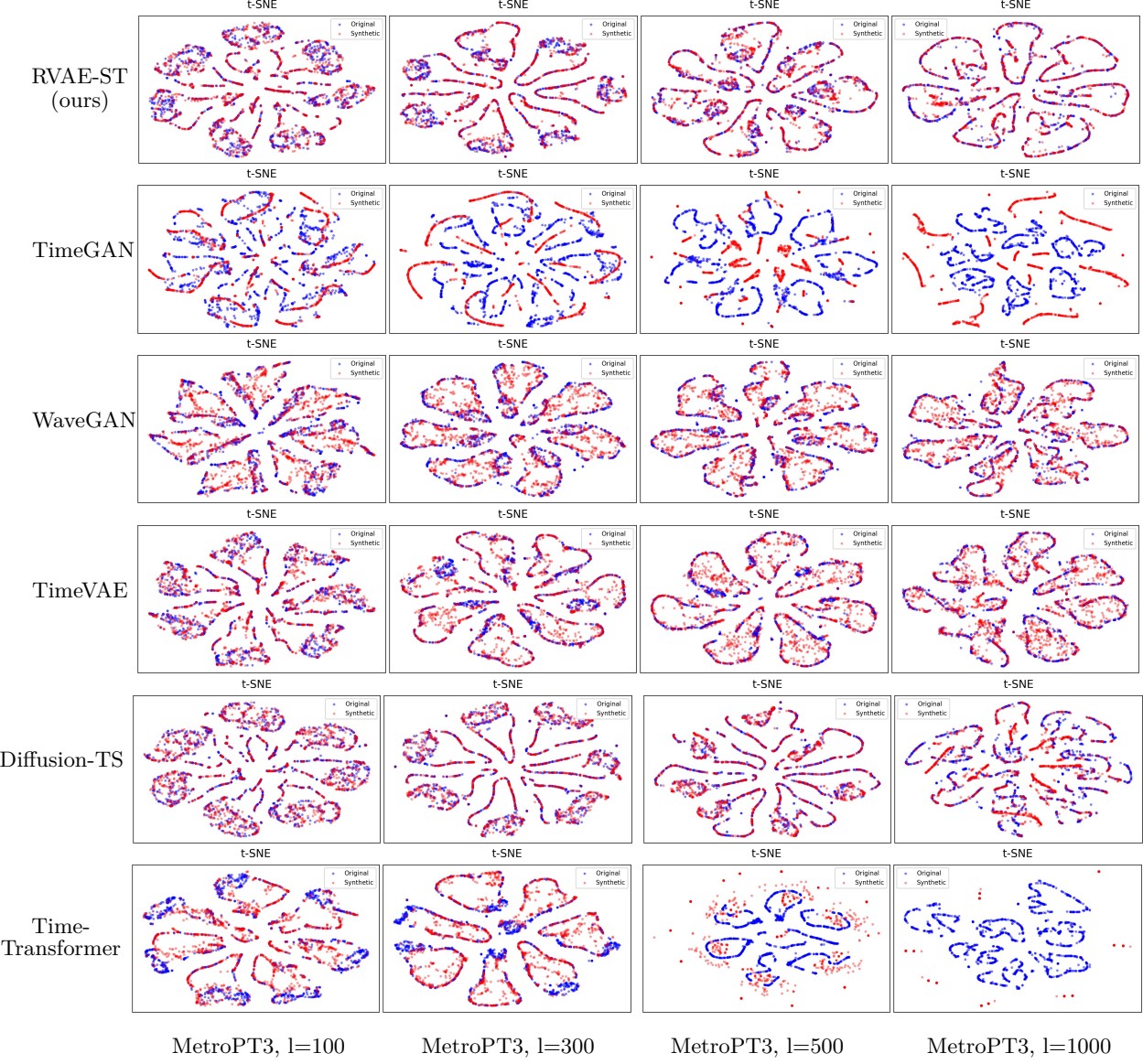

Figure 17: t-SNE plots for all sequence lengths on the MetroPT3 dataset. RVAE-ST and Diffusion-TS perform the best across all sequence lengths, with RVAE-ST slightly outperforming at $l = 1000$. TimeVAE and WaveGAN perform similarly, but exhibit more outliers in the plots. TimeGAN and Time-Transformer perform similarly to the PCA results, showing significant degradation as the sequence length increases.

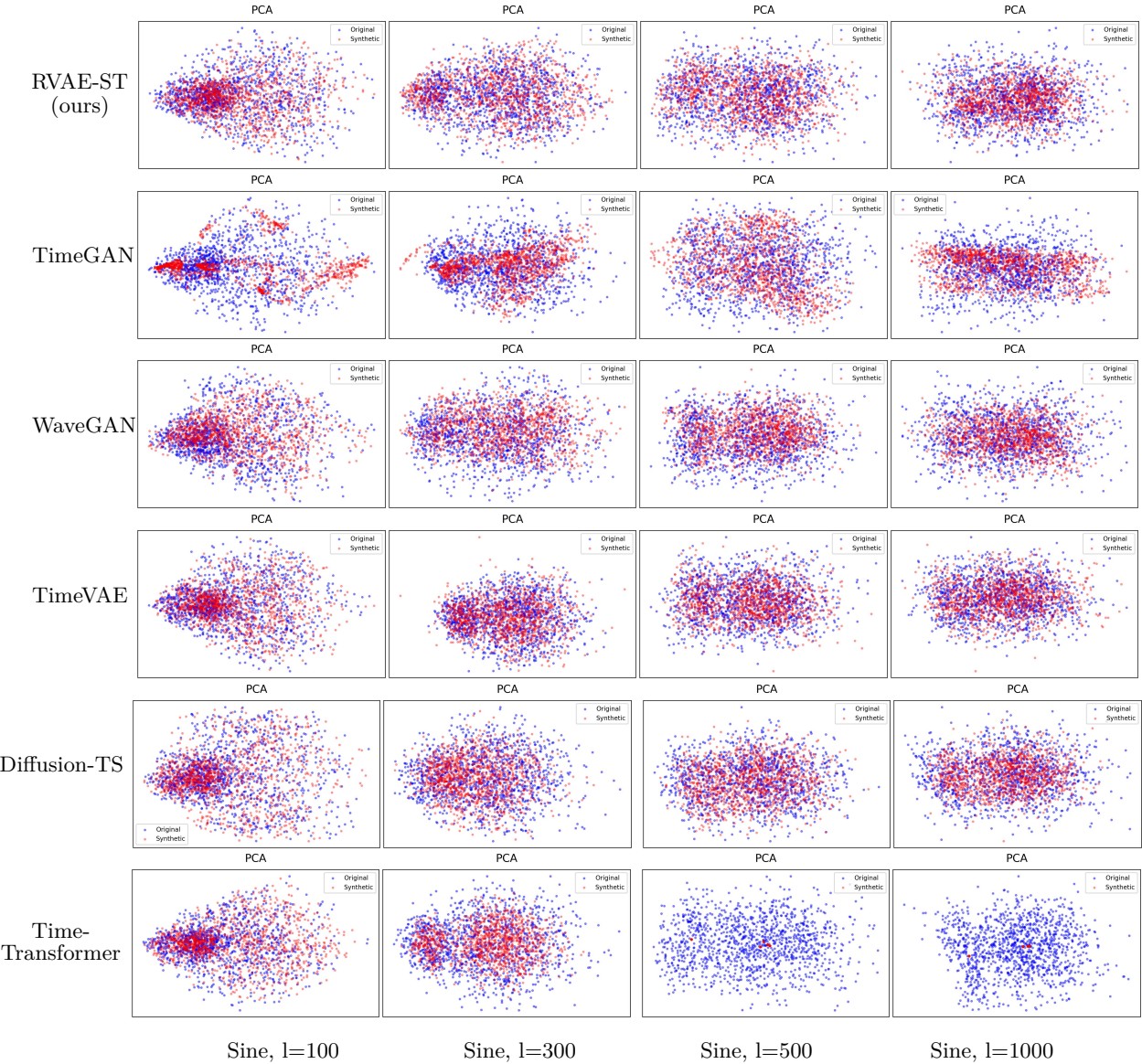

Figure 18: PCA plots for all sequence lengths on the Sine dataset. At $l = 100$, all models perform similarly well, except TimeGAN which consistently performs less effectively. From $l = 500$ onward, Time-Transformer also shows a decline in performance. RVAE-ST, Diffusion-TS, WaveGAN and TimeVAE perform equally well throughout all sequence lengths. However, when looking at Figure 3, this does not fully reflect the models performance, as the limitations in accounting for temporal dependencies lead to significantly reduced effectiveness.

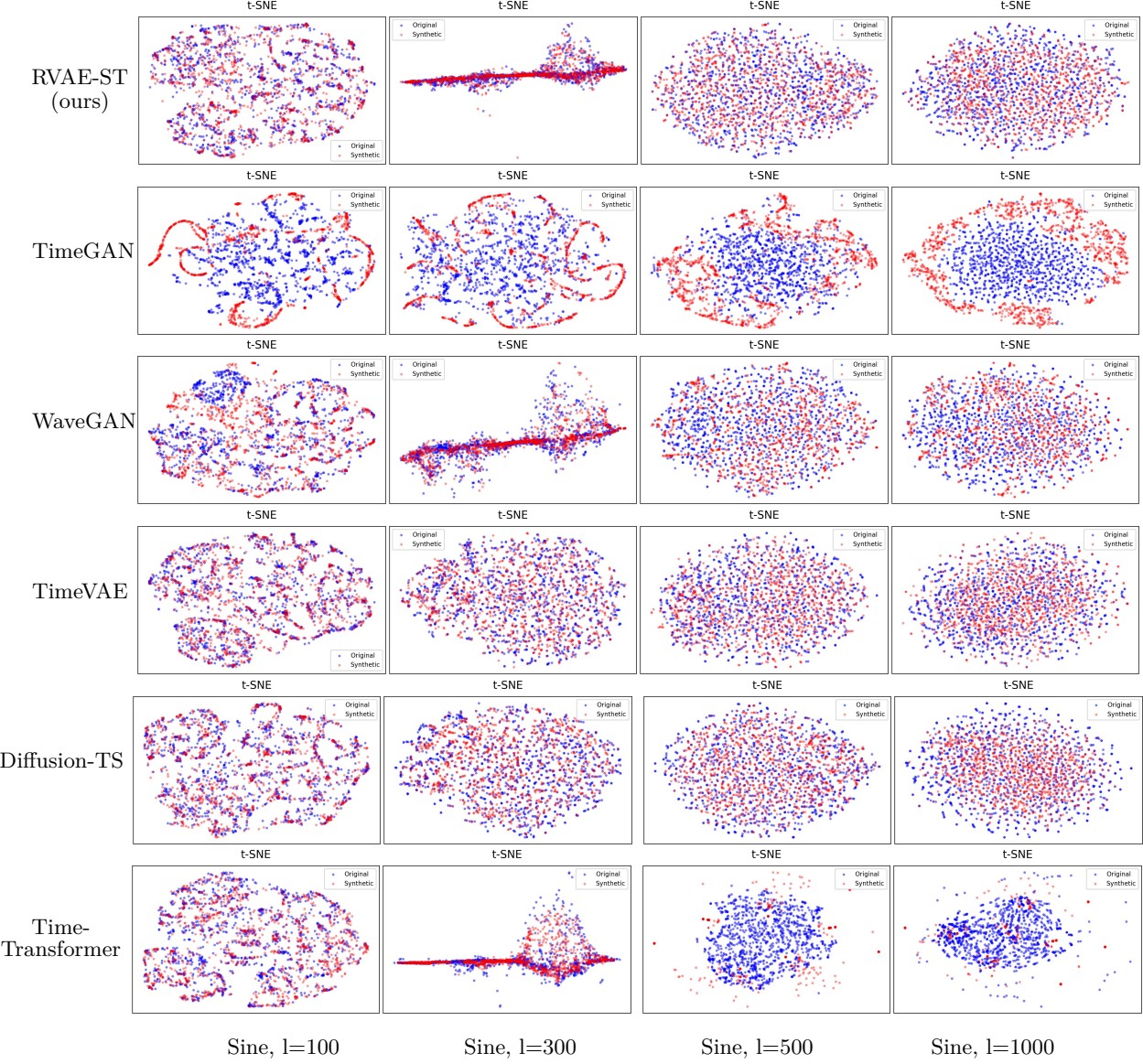

Figure 19: t-SNE plots for all sequence lengths on the Sine dataset. At $l = 100$, all models perform similarly well, except TimeGAN which consistently performs less effectively. From $l = 500$ onward, Time-Transformer also shows a decline in performance. RVAE-ST, Diffusion-TS, WaveGAN and TimeVAE perform equally well throughout all sequence lengths. However, when looking at Figure 3, this does not fully reflect the models performance, as the limitations in accounting for temporal dependencies lead to significantly reduced effectiveness.

