# OpenReview forum: "Generative Models for Long Time Series: Approximately Equivariant Recurrent Network Structures for an Adjusted Training Scheme"
_TMLR — Rejected by TMLR_

### Review · Reviewer_c7fZ · 2025-02-18

**Summary Of Contributions:**

The paper presents a training scheme for variable-lenght VAE-based sequence-processing deep architectures.

**Audience:**

No

**Broader Impact Concerns:**

Nothing to report.

**Claims And Evidence:**

No

**Requested Changes:**

- In page 4 you state "assuming that all variables represent probabilities rather than samples", but it is not clear what this means.
- In page 4: what is a time-distributed linear layer?
- When motivating your research in Section 3.1, you state that "transformers lack the ability to capture temporal dynamics as effectively as recurrent layers". Could you please provide references about that?

**Strengths And Weaknesses:**

The paper is clearly written, and there are experiments in three datasets.

However, there are a set of points that raise significant concern.

One very important issue of the paper is the lack of positionning with respect to the state-of-the-art. The paper proposes a VAE-based architecture that can deal with sequences. This is exactly the framework of DVAE (dynamical VAE), proposed by Girin et al in 2021. Beyond this reference in particular, the submitted article neglects all the work done since 2014 forward combining VAE and recurrent architectures. Well-known models include SRNN, STORN, VRNN, RVAE (which shares part of the name with the proposed method and is not cited!!!), DKF, etc. The lack of positioning with respect to this very large body of literature is not acceptable.

Regarding the model itself, the description is very brief, and no justification or motivation is given to why only one latent variable is modelled, instead of a sequence of latent variables -- which would invuitively be much more powerful. Hierarchical models (e.g. a sequence of latent variables combined with a single latent variable on top) are also possible and not discussed. Would the proposed training scheme be valid also here?

Regarding the architecture implementing the model, the reader lacks intuition of why LSTM layers are stacked, of even used (why not GRU or standard RNN). If the objective is to present a training scheme, all these choices seem possible and comparable.

The authors claim that "the combination of RNNs with time-distributed layers [...] has not been widely
emphasized in the literature". Since there is not specific reference provided for "Time-Distributed Layers", I would assume that it is a layer applied to all time steps equally (which aligns with the fact that the model size does not grow with the sequence length as claimed by the authors). If this is the case, then ALL RNNs, LSTMs, GRUs, and DVAEs, have time-distributed layers. This is very puzzling.

The training scheme proposed in the paper (which seems to be the main contribution of the manuscript) consists on starting with small chunks, and progressively increasing the sequence length. While this might prove effective, I do not know if it constitutes a scientific contribution by itself. The probability-based motivation is not really compelling, since it does not tell us how good/bad an approximation it is. In addition, no general recommendation for initial/increment sequence length values are provided. All in all, it is very difficult to extract new knowledge from the paper, at the methodological level.

Regarding the metrics, I am concerned that the only metric that evaluates the quality of the time-dependencies is the discriminative score (the ELBO as well, but it cannot be used with all methods), since the PCA and t-SNE operate independently per frame. Even the discriminative score is a classification score that showcases whether or not the temporal patterns or the generated data are correctly aligned to the ground truth ones. But I am not convinced that a higher discriminative score implies better representation of the time-dependencies.

---

> ### Author Response · Authors · 2025-04-16
>
> Thank you for your valuable feedback and for introducing the dVAE framework. Your insights allowed us to better position our work in relation to existing methods and to clearly highlight the distinctions between our approach and dVAE. This was an important aspect of the revision, and we appreciate your contribution to improving the clarity and context of our paper.
>
> ***One very important issue of the paper is the lack of positionning with respect to the state-of-the-art. The paper proposes a VAE-based architecture that can deal with sequences. This is exactly the framework of DVAE (dynamical VAE), proposed by Girin et al in 2021. Beyond this reference in particular, the submitted article neglects all the work done since 2014 forward combining VAE and recurrent architectures. Well-known models include SRNN, STORN, VRNN, RVAE (which shares part of the name with the proposed method and is not cited!!!), DKF, etc. The lack of positioning with respect to this very large body of literature is not acceptable.***
>
> Thank you for your valuable feedback regarding the positioning of our work within the broader literature on VAE-based time series models.
>
> While our model shares the VAE-based structure, it does not follow the framework of dynamic VAEs (dVAEs) as introduced by Girin et al. (2021). dVAEs typically employ a sequence of latent variables to model evolving temporal dynamics. In contrast, our approach uses a single global latent vector for the entire sequence, focusing on approximate translational equivariance and stationarity rather than temporal flexibility. We now clarify this distinction explicitly in the revised Related Work section.
>
> Regarding earlier works such as SRNN, STORN, VRNN, DKF, RVAE and others—these are comprehensively surveyed in the dVAE paper by Girin et al. (2021), which we now cite and discuss. Since our model departs fundamentally from this class of approaches, we refer to the survey paper rather than listing each individual model.
>
> We would like to clarify that while we build upon the Recurrent Variational Autoencoder (RVAE) by Fabius and van Amersfoort (2014), there is another RVAE framework discussed in the dVAE paper by Simon et al. (2020). To avoid confusion, we explicitly cite the Fabius et al. (2014) paper as the source of our model and clarify the distinction in the newly added ***Related Work*** section.
>
> We sincerely acknowledge that the citation to the original RVAE paper by Fabius and van Amersfoort (2014) was mistakenly omitted in the initial submission. This has now been corrected in the updated version, where our model’s connection to RVAE and our modifications are clearly described.
>
>
> ***Regarding the model itself, the description is very brief, and no justification or motivation is given to why only one latent variable is modelled, instead of a sequence of latent variables -- which would invuitively be much more powerful. Hierarchical models (e.g. a sequence of latent variables combined with a single latent variable on top) are also possible and not discussed. Would the proposed training scheme be valid also here?***
>
> We deliberately use a single global latent variable to capture global sequence properties, while the decoder LSTM models local temporal dynamics. This design reflects our inductive bias: approximate translational equivariance and stationarity, which are central to our modeling goals.
>
> While modeling a sequence of latent variables (as in dynamic or hierarchical VAEs) offers more flexibility, it also introduces complexity and weakens stationarity assumptions. Additionally, such models are often affected by posterior collapse—a phenomenon that also occurs in our model. However, in our case, posterior collapse is not a defect but rather a natural consequence of the model’s design: the latent variable encodes global information, and the decoder learns to generate locally consistent sequences under stationarity.
>
> We emphasize this design choice throughout the paper, and revisit it in the ***Related Work*** section, where we contrast our approach with dynamic VAEs and justify our use of a single latent vector in alignment with our inductive biases.
>
> ***Regarding the architecture implementing the model, the reader lacks intuition of why LSTM layers are stacked, of even used (why not GRU or standard RNN). If the objective is to present a training scheme, all these choices seem possible and comparable.***
>
> Thank you for this observation. Stacking LSTM layers is a standard design choice in sequence modeling and was found to work well in our setting. While our training scheme is compatible with other recurrent architectures such as GRUs or standard RNNs, we chose LSTMs based on preliminary experiments, where they performed slightly better than GRUs.

---

> ### Author Response · Authors · 2025-04-16
>
> ***The authors claim that "the combination of RNNs with time-distributed layers [...] has not been widely emphasized in the literature". Since there is not specific reference provided for "Time-Distributed Layers", I would assume that it is a layer applied to all time steps equally (which aligns with the fact that the model size does not grow with the sequence length as claimed by the authors). If this is the case, then ALL RNNs, LSTMs, GRUs, and DVAEs, have time-distributed layers. This is very puzzling.***
>
> Of course, RNNs apply the same transition function across time steps, which can be interpreted as a form of time-distributed computation. However, as seen in the original RVAE implementation by Fabius and van Amersfoort (2014) https://github.com/arunesh-mittal/VariationalRecurrentAutoEncoder/blob/master/vrae.py, lines 169–179}, this is not always followed by an explicit time-distributed layer in the decoder to construct the final output sequence.
>
> ***The training scheme proposed in the paper (which seems to be the main contribution of the manuscript) consists on starting with small chunks, and progressively increasing the sequence length. While this might prove effective, I do not know if it constitutes a scientific contribution by itself. The probability-based motivation is not really compelling, since it does not tell us how good/bad an approximation it is. In addition, no general recommendation for initial/increment sequence length values are provided. All in all, it is very difficult to extract new knowledge from the paper, at the methodological level.***
>
> Thank you for your feedback. We understand your concerns, and we would like to clarify that the main contribution of our work lies in the novel combination of inductive biases, network topology, and training scheme within the recurrent variational autoencoder (RVAE) architecture. The training scheme is just one of several components that together constitute the novelty of our approach.
>
> As we mention in the paper, the choice of initial and increment sequence lengths was somewhat arbitrary. However, our experiments demonstrate that this approach leads to significant improvements, particularly in enabling the model to learn effectively from long sequences. We acknowledge that the selection of increment values remains an open research question, and we believe that further investigation into optimal values could lead to even better results.
>
> ***Regarding the metrics, I am concerned that the only metric that evaluates the quality of the time-dependencies is the discriminative score (the ELBO as well, but it cannot be used with all methods), since the PCA and t-SNE operate independently per frame. Even the discriminative score is a classification score that showcases whether or not the temporal patterns or the generated data are correctly aligned to the ground truth ones. But I am not convinced that a higher discriminative score implies better representation of the time-dependencies.***
>
> Thank you for your concern regarding the evaluation of temporal dependencies. In response, we have added the Contextual FID score to our evaluation and we have further addressed this point in the main rebuttal on ***Contextual FID score***.

---

### Review · Reviewer_1ytc · 2025-02-22

**Summary Of Contributions:**

This paper proposes an RNN-based VAE and a progressive training scheme for long time series generation. RNN is selected as the backbone due to its approximately equivariant bias to handle (semi-)stationary and quasi-periodic time series. The progressive training scheme, which gradually increases the input sequence length during training, is proposed to address the difficulty recurrent layers have in handling long sequences. The qualitative and quantitative metrics across three datasets demonstrate the superiority of the proposed model over four baseline models.

**Audience:**

No

**Broader Impact Concerns:**

Not applied.

**Claims And Evidence:**

No

**Requested Changes:**

1. Please polish the writing to make the work more rigorous. It is recommanded to use more formal equations instead of descriptional texts.
2. Please compare with more recent models, especially diffusion-based ones.
3. See my other questions in the weaknesses section.

**Strengths And Weaknesses:**

## Strengths
1. The proposed VAE and corresponding training scheme are intuitive and easy to understand.
2.  The evaluation using multiple metrics and visualizations is comprehensive, demonstrating the effectiveness of the proposed approach.

## Weaknesses
1. The writing is not very rigorous, employing a significant amount of vague descriptions. For example:
   - the definition of time-shift invariance in Section 3.1 is confusing, especially with the Equation $p_\theta(x) = p_\theta(\tau(x))$. What's the formulation of the operator $\tau(x)$ when both $x, \tau(x) \in R^l$. Considering in the experiment, the samples are generated by shifting the sliding window on a very long sequence, does $p_\theta(x) = p_\theta(\tau(x))$ mean the probability of generating each sample is equal?
   - In the structure selection part, many unfounded claims are utilized. Why convolutional layers alone are insufficient for generating sequences of variable lengths from fixed-length inputs? What does it mean by RNNs are only approximately equivariant? If so, why to use approximately equivariant structures instead of fully equivariant ones? Why do Transformers lack the ability to capture temporal dynamics, considering they are now the fundament of NLP where position information is extremely important, and Transformers are also widely used in plenty of time series models?

2. The proposed approaches lack novelty. This paper merely presents a specific implementation of VAE, yet it fails to provide a convincing explanation as to why VAE is employed or why this particular implementation was chosen.

3. The compared baseline models are out-of-date as all of them are proposed before 2021. Please compare with more recent models, especially diffusion-based ones[1,2].

[1] Diffusion-TS: Interpretable Diffusion for General Time Series Generation.

[2] On the Constrained Time-Series Generation Problem.

---

> ### Author Response · Authors · 2025-04-16
>
> Thank you for your insightful feedback, which especially prompted us to clarify several important aspects in the section on equivariance. Your suggestions were instrumental in refining our explanation, allowing us to present this key concept with greater precision and clarity, ultimately enhancing the paper.
>
> ***1.1***
> Thank you for your comment. We agree that the original presentation of the time-shift operator and the approximate invariance condition lacked clarity. We have revised this section to clarify both the meaning of the operator $\tau(x)$ and the interpretation of the expression $p_\theta(x) \approx p_\theta(\tau(x))$.
>
> More specifically, we now explicitly define $x$ as a function of time, $x:\mathbb{Z} \to \mathbb{R}^c$, and the time-shift operator $\tau$ as acting on such functions via $\tau(x)(t) = x(t+1)$. The revised formulation emphasizes that our objective is to learn a model that assigns approximately the same probability to a sequence and its time-shifted version. In other words, the likelihood of generating a particular temporal pattern should be invariant to the absolute time index at which it occurs—an inductive bias that is particularly beneficial for stationary and quasi-periodic time series.
>
> We also clarified that, in practice, the model is trained on finite-length windows (e.g., $x \in \mathbb{R}^{\ell \times c}$), and that the distribution is learned over these windows. The approximate invariance condition then expresses that similar short patterns are likely to occur throughout the timeline, and the model reflects this by learning similar likelihoods for those shifted windows.
>
> These changes are now reflected in Section 4.1 of the revised manuscript.
>
> ***Why convolutional layers alone are insufficient for generating sequences of variable lengths from fixed-length inputs?***
> Convolutional layers alone are insufficient for generating sequences of variable lengths from fixed-length inputs because they typically operate within a fixed receptive field and are designed to enhance resolution within a given time window rather than extend the sequence itself. In time series, this corresponds to refining local details rather than generating temporally longer outputs. For instance, PSA-GAN [s1] uses progressive growing to increase temporal resolution, but the overall sequence length remains fixed.
> We have now made that clear in Section ***4.1***.
>
> [s1] Paul, Jeha, et al. "Psa-gan: Progressive self attention gans for synthetic time series." _arXiv preprint arXiv:2108.00981_ (2021).
>
> ***What does it mean by RNNs are only approximately equivariant?***
> Thank you for your feedback. We have addressed this point in the main rebuttal on ***equivariance***.
>
> ***If so, why to use approximately equivariant structures instead of fully equivariant ones?***
> Our goal was to design a model where the number of trainable parameters remains independent of the sequence length. While full equivariance is theoretically desirable, it does not automatically ensure this practical property. In contrast, recurrent architectures like LSTMs offer a form of approximate equivariance through shared weights across time steps, while also enabling parameter efficiency and variable-length generation. For our use case, this trade-off was more favorable than enforcing strict equivariance.

---

> ### Author Response · Authors · 2025-04-16
>
> ***Why do Transformers lack the ability to capture temporal dynamics, considering they are now the fundament of NLP where position information is extremely important, and Transformers are also widely used in plenty of time series models?***
> Thank you for your feedback. We have addressed this point in the main rebuttal on ***On Transformer temporal dynamics***.
>
> ***The proposed approaches lack novelty. This paper merely presents a specific implementation of VAE, yet it fails to provide a convincing explanation as to why VAE is employed or why this particular implementation was chosen.***
> While we agree that Variational Autoencoders (VAEs), repeat vectors or time-distributed layers  are not novel by themselves, our approach is not merely a generic implementation of a VAE. Among the primary approaches for generative modeling of time series, VAEs stand out for their simplicity and direct approach to probabilistic modeling. In contrast to more complex frameworks like Diffusion Models, VAEs are computationally efficient, easier to implement, and well-suited for modeling the underlying data distribution of time series. This makes them particularly attractive for practical applications, where computational resources and time may be limited.
>
> However, the novelty of our approach lies not in the use of these components themselves, but in how we combine them with inductive biases, network topology, and training strategies specifically designed for time series data. By leveraging these elements, our model effectively captures temporal dependencies and maintains high performance across different datasets. Our experiments now show that this novel combination leads to superior performance compared to state-of-the-art models such as Diffusion-TS.
>
> ***The compared baseline models are out-of-date as all of them are proposed before 2021. Please compare with more recent models, especially diffusion-based ones***
> Thank you for your feedback.  We have addressed this point in the main rebuttal on  ***Two additional baseline models***. This suggestion has particularly added value to our paper.

---

### Review · Reviewer_BfZt · 2025-03-26

**Summary Of Contributions:**

The authors proposed a Variational Autoencoder (VAE) model with Recurrent layers to generate synthetic time series of different lengths. Given a time series as input, their model generates synthetic time series from approximately the same distribution. The authors note that convolutional neural networks are likely to scale well with the length of the input sequence, but cannot produce outputs of varying lengths. On the other hand, LSTMs can model increasingly long sequences, but are only approximately equivariant. The authors use these ideas to create an model which can model stationary time series well.

To ensure that their model scales with the length of the input, the authors propose a training mechanism where they train models on time series with gradually increasing lengths until convergence. They find that their proposed training mechanism, performs better than naively training their model on sequences of different lengths in arbitrary order.

**Audience:**

Yes

**Broader Impact Concerns:**

I do not believe that there are any ethical concerns that stem from the paper and its methods.

**Claims And Evidence:**

Yes

**Requested Changes:**

Please review the previous section.

Addressing points marked as **major** are likely to improve my assessment of the paper the most.

**Strengths And Weaknesses:**

### Strengths
1. The paper is well written. The problem is important, and the methods are simple but performant.

### Claims & Weaknesses
1. **Equivariance, Time-shift Invariance & Stationarity:** I would like to better understand the need to design models which can model stationary time series well. Why does the designed model have to be equivariant to perform well in modeling stationary time series? A lot of time series models are likely not equivariant and do just fine modeling stationary data.
2. **Regarding Transformers:** The authors claim that "Transformers lack the ability to capture temporal dynamics as effectively as recurrent layers." Do you have any evidence for this statement? Most state-of-the-art time series foundation models as well as deep forecasting models are either Transformer-based, or stacked MLPs, and not RNNs.
3. **Time-distributed Linear Layers:** The authors claims that "Although the combination of RNNs with time-distributed layers is not entirely novel, it has not been widely emphasized in the literature. We offer a fresh perspective by focusing on its application to stationary data." I agree that this is not novel. Time distributed linear layers are widely used in time series modeling, and I am not sure what the authors mean by it being "not widely emphasized". What is the fresh perspective that you are adding?
4. **Equivariance:** The authors claim that "Equivariance is maintained throughout the network.". Could you provide evidence for why this is the case, when LSTMs are only "approximately equivariant". Also, what does approximately invariant mean?
5. **Length Generalization Training:** The idea of increasing the length of sequences gradually during training is simple and effective. I wonder if it is novel. It seems it is used in the NLP setting, see [1] as an example. I wonder how does your training mechanism deal with length generalization i.e. can it model time series shorter and longer than it has seen during training?
6. **Multivariate Modeling:** How does RVAE-ST model multivariate time series?
7. **Baselines:** This is a **major** issue. The authors evaluated their method against baselines from 2017--2021. Many new model have been proposed, and it would be important to compare against them. Please look at 2 -- 4 for example and the references therein.
8. **Datasets:** This is a **major** issue. The authors use very few datasets (only 3). I would recommend evaluating your method on more datasets from diverse domains, with diverse characteristics (non-periodic, non-stationary for example).
9. **Stationarity:** How would the model perform when time series is not stationary?
10. **Evaluation:** I wonder what is the value of generating synthetic time series. If it is to improve the accuracy of models on downstream task, then I believe for some tasks, improvement in downstream tasks can be reported. In addition, it seems 2 and 3 report other metrics based on Frechet Inception Distance. It might be a good idea to report these metrics in your work, or justify the ones that you have used.
11. **Related Work:** This is a **major** issue. The authors also do not have a related work section. I would encourage adding this section, organize it into themes, and position their work against prior work on related topics.

### References
1. Tworkowski, Szymon, et al. "Focused transformer: Contrastive training for context scaling." Advances in neural information processing systems 36 (2023): 42661-42688.
2. Yuan, Xinyu, and Yan Qiao. "Diffusion-ts: Interpretable diffusion for general time series generation." arXiv preprint arXiv:2403.01742 (2024).
3. Narasimhan, Sai Shankar, et al. "Time weaver: A conditional time series generation model." arXiv preprint arXiv:2403.02682 (2024).
4. Liu, Yuansan, et al. "Time-transformer: Integrating local and global features for better time series generation." Proceedings of the 2024 SIAM International Conference on Data Mining (SDM). Society for Industrial and Applied Mathematics, 2024.

---

> ### Author Response · Authors · 2025-04-16
>
> We sincerely thank you for your insightful and constructive review. Many of your points had a direct impact on the revision and have noticeably strengthened the paper both in clarity and substance.
>
> ***1***
> The key idea behind our approach is that stationary time series exhibit consistent patterns over time. Equivariance helps the model generalize better across time shifts, which is essential for capturing the underlying structure in stationary data, especially on long sequences. While LSTM layers contribute approximately to equivariance, the rest of the network’s structure is fully equivariant. This, combined with the ability to subsequently increase sequence lengths during training, enables our model to learn effectively from long time series combined with the training schema.
>
> While it is true that many time series models are not explicitly equivariant and still perform reasonably well on stationary data, they do not actively exploit the structure of stationarity. Our approach shows that leveraging this property through equivariance leads to more efficient and scalable learning, as confirmed by our experiments on synthetic and real-world data.
>
> ***2***
> Thank you for your feedback. We have addressed this point in the main rebuttal on ***On Transformer temporal dynamics***
>
> ***3***
> Thank you for your feedback. We have addressed this point in the main rebuttal on ***Deeper Model explenation***.
>
> ***4***
> Thank you for your feedback. We have addressed this point in the main rebuttal on  ***equivariance***.
>
> ***5***
> Thank you for your feedback. We have addressed this point in the main rebuttal on  ***Extended Sequences***.
>
> ***6***
> Our model handles multivariate time series by learning a joint latent representation of the temporal dynamics and features across all variables. The latent space does not directly correspond to the number of features or sequence length; rather, it captures the underlying patterns and dependencies within the data. This allows the model to generalize well to long time series and multivariate data, even if the dimensionality of the latent space differs from the input data’s structure.
>
> ***7***
> Thank you for your feedback. We added [2] and [4] to the baseline models. [3] did not publish its code. We have addressed this point in the main rebuttal on ***Two additional baseline models***. This suggestion has particularly added value to our paper.
>
>
> ***8***
> Thank you for your feedback. We have addressed this point in the main rebuttal on ***Two additional datasets***. This suggestion has particularly added value to our paper.
>
>
> ***9***
> To address how the model performs with non-stationary time series, we refer to the MetroPT3 dataset. While this dataset contains recurring patterns, it is one of the less stationary datasets in our study, as the frequency of these patterns varies significantly, and some signals occasionally drop out completely. Our results on the MetroPT3 dataset are comparable to [2] and TimeVAE, but we do not outperform them.
>
> ***10***
> Thank you for your feedback. We have addressed this point in the main rebuttal on ***Contextual FID score***. This suggestion has particularly added value to our paper.
>
> ***11***
> Thank you for your feedback. We have addressed this point in the main rebuttal on ***Related Work***. This suggestion has particularly added value to our paper.

---

### Author Response · Authors · 2025-04-16

- ***On Transformer temporal dynamics***
  We do not claim that Transformers cannot model temporal dynamics. Clearly, they have been extremely successful in NLP and, more recently, in time series modeling as well. Our point is that their inductive bias is more naturally aligned with tasks involving flexible word order and long-range semantic dependencies, such as in NLP. In contrast, time series data can benefit from strong local continuity and strict temporal order, which are directly encoded in the dynamics of recurrent models.

  We have also revised the original phrasing that suggested Transformers “lack the ability” to model temporal dynamics, as we acknowledge that this wording did not accurately reflect our intended nuance. We now emphasize this distinction more carefully in the revised text and do not question the expressive power of Transformers per se, but rather highlight how their inductive bias may not be as naturally suited to the rigid sequential structure of time series data compared to recurrent models.

  Additionally, we expanded on the practical limitations of Transformer-based architectures for time series applications. Since the self-attention mechanism scales quadratically with sequence length, models trained on longer sequences quickly become resource-intensive. In our case, the comparison models _Diffusion-TS_ and _Time-Transformer_ exceeded 20 GB of GPU memory at a sequence length of $l = 1000.$

- ***RCGAN removed***
  We removed RCGAN from the baseline comparison, as it turned out to be outdated and unstable in our setting, offering no meaningful performance benefit or additional insight.

- ***Changes in Blue***
  All major changes are marked in blue in the revised version to facilitate review.

---

### Author Response · Authors · 2025-04-16

- ***Novelty***
  Regarding the novelty of our contribution, we clarified that it does not rely on the training scheme alone, but arises from a combination of inductive biases, model architecture, and training strategy. Specifically, our architecture integrates approximate time-translation equivariance into a recurrent VAE framework while keeping the number of parameters fixed, independent of sequence length.

- ***Equivariance***
  Since the question of equivariance appeared in multiple reviews, we now explain this concept in more detail. We highlight the distinction between true equivariance and the approximate shift-equivariance exhibited by RNNs in the presence of hidden state initializations and finite context windows. Basically, RNNs are only approximately equivariant because their hidden states depend on initialization and finite context, especially at the beginning of a sequence. As a result, small shifts in the input may lead to different outputs early on. However, this effect diminishes over time, and for long sequences, shift equivariance holds approximately.

  We highlight the new relevant explanation from the paper:

> To illustrate this further, consider the case where we use an LSTM cell $(y_i,h_{i+1},c_{i+1})=f(x_i,h_i,c_i)$ mapping input $x_i$, hidden state $h_i$, and cell state $c_i$ to the output $y_i$. We will denote the map producing the hidden and cell state by $(h_{i+1},c_{i+1})=\hat f(x_i,h_i,c_i)$. Now consider mapping two overlapping time series $X=[x_0,\ldots,x_{n-1}]$ and $X'=[x_1,\ldots,x_n]$ via this LSTM-cell. Of course, in general $f(x_0,h,c)\neq f(x_1,h,c)$ for initializations $h$ and $c$ of hidden and cell state. However, for long time series (i.e. $n\gg 0$) we can usually expect convergence in hidden and cell state over time, more precisely:
> $\hat f(x_k,\hat f(x_{k-1},\hat f(x_{k-2},\hat f(\ldots,\hat f(x_0,h,c)\ldots)))) \approx \hat f(x_k,\hat f(x_{k-1},\hat f(x_{k-2},\hat f(\ldots,\hat f(x_1,h,c)\ldots))))$
> i.e. shifting a long time series will be approximately equivariant.

- ***Extended Sequences***
  One reviewer raised the question of how our model behaves when the sequence length is increased at inference time while keeping the weights fixed. In response, we added a dedicated section in the appendix analyzing this setting. The results support our intuition about the model’s inductive bias toward approximate translational equivariance and demonstrate its ability to generalize effectively to longer horizons. We believe this addition provides further valuable insights.

- ***Deeper Model explenation***
  Several reviewers noted uncertainty regarding specific components of our model, such as the use of a single global latent vector and the role of the time-distributed layer. To address this, we revised Sections 4.1 and 4.2 to clarify our design choices and motivations. In particular, we explain how the time-distributed linear layer supports equivariance and why we model a global latent representation instead of a latent sequence.

  We highlight the new relevant explanation from the paper:

> Next, the generative network $p_{\theta}(x|z)$ reconstructs the data from the latent variable $z$. To achieve this, the latent variable $z$ is repeated across all time steps (using a repeat vector), ensuring that $z$ remains constant at each time step and is shared throughout the entire sequence. Mathematically, this can be expressed as:
>
> $z_t = z \quad \text{for all} \, t \in \{1, 2, \dots, n\}$
>
> where $n$ denotes the total number of time steps in the sequence.
> The repeat vector is followed by stacked LSTM layers. Finally, a time-distributed linear layer is applied in the output. This layer operates independently at each time step, applying the same linear transformation to the LSTM output at every time step, which can be viewed as a $1 \times 1$ convolution across the time dimension, with shared weights across all time steps.
>
> The time-distributed layer is inherently equivariant with respect to time-translation, preserving temporal structure and shifts over time. Together with our LSTM-based approach and the repeat-vector mechanism, this design ensures that the number of trainable parameters remains independent of the sequence length, while also enabling an adapted training scheme that can accommodate increasing sequence lengths.

---

### Author Response · Authors · 2025-04-16

We would like to sincerely thank all reviewers and the Action Editor for their time, thoughtful feedback, and constructive suggestions. We carefully revised our manuscript in response to the comments and outline the most important changes and clarifications below:

- ***Related Work***
  We have added a related work section that highlights key models for time series generation, including GANs, VAEs, diffusion models, and transformer-based approaches. We clarify that our approach builds upon the Recurrent Variational Autoencoder (RVAE) by Fabius and van Amersfoort (2014) and describe the enhancements we have made. Additionally, we distinguish our method from dynamic VAEs (dVAE) by utilizing a fixed latent vector across the entire sequence, which emphasizes translational equivariance and stationarity, while dVAEs rely on a sequence of latent variables to allow for greater flexibility.

- ***Two additional datasets***
  We have added two additional datasets to our evaluation. The first is the Synthetic Sine dataset, commonly used in time series research to assess a model’s ability to generate periodic signals. The second is MetroPT3, a multivariate sensor dataset collected from analogue and digital sensors installed on a compressor. While it contains recurring patterns, it is considerably less stationary than the other datasets. Its non-stationarity stems from varying pattern frequencies, anomalies and occasional signal dropouts, posing additional challenges for generative models. The results on these datasets further reinforce our intuition about the importance of stationarity and highlight the inductive biases of our model.

- ***Two additional baseline models***
  We integrated two very recent models, Time-Transformer (Yuan & Qiao, 2024) and Diffusion-TS (Liu et al., 2024). Our model outperformed Time-Transformer across all datasets and achieved notably strong results compared to Diffusion-TS. A more detailed breakdown is provided in the next point.

- ***Contextual FID score***
  To strengthen the evaluation, we included the Contextual FID score as an additional metric. This score directly assesses the quality of generated time series with respect to temporal consistency and global fidelity. It has become the most compelling indicator of our model’s generative capabilities and supports our main claims. The following table presents a subset of the Contextual FID scores for the five datasets at sequence lengths of 500 and 1000, extracted from the full table in the paper. We compare the performance of our model (RAVE-ST) with TimeVAE and Diffusion-TS, as these models represent the strongest baselines. Notably, our model demonstrates significantly better performance on the more stationary datasets (Electric Motor, ECG, Sine), while we remain comparable on the less stationary datasets (ETT, MetroPT3). The lower the score, the better:

| Dataset            | Model              | 500           | 1000          |
| ------------------ | ------------------ | ------------- | ------------- |
| **Electric Motor** | **RVAE-ST (ours)** | **0.10±0.01** | **0.24±0.02** |
|                    | TimeVAE            | 1.06±0.14     | 1.19±0.09     |
|                    | Diffusion-TS       | 1.10±0.11     | 1.93±0.13     |
| **ECG**            | **RVAE-ST (ours)** | **0.14±0.02** | **0.46±0.06** |
|                    | TimeVAE            | 1.07±0.10     | 1.30±0.08     |
|                    | Diffusion-TS       | 0.52±0.03     | 3.74±0.22     |
| **ETT**            | **RVAE-ST (ours)** | **0.79±0.07** | 1.82±0.16     |
|                    | TimeVAE            | 0.97±0.10     | **1.56±0.14** |
|                    | Diffusion-TS       | 2.16±0.17     | 2.55±0.27     |
| **Sine**           | **RVAE-ST (ours)** | **0.46±0.03** | **0.42±0.03** |
|                    | TimeVAE            | 1.26±0.14     | 3.03±1.00     |
|                    | Diffusion-TS       | 0.74±0.04     | 2.66±0.20     |
| **MetroPT3**       | **RVAE-ST (ours)** | 2.81±0.37     | 2.84±0.22     |
|                    | TimeVAE            | 2.02±0.29     | **2.08±0.31** |
|                    | Diffusion-TS       | **1.82±0.09** | 6.97±0.75     |

---

### Decision · Action_Editor_cdBA · 2025-06-30

**Recommendation:** Reject

**Additional Comments:**

Please consider the comments made by the reviewers and myself in your revision -- having a clear definition of "time-shift equivariance" is needed, why is this concept different from stationarity?

Also I suggest an experiment verifying the "approximate equivariance" assumption in the trained models, and perform some ablation studies here. This is to check whether the experimental improvements do come from the "approximate time-shift equivariance" assumption.

**Audience:**

Yes

**Audience Explanation:**

A paper for generative modelling applied to time series data, should be interesting to the time series modelling community.

**Claims And Evidence:**

No

**Claims Explanation:**

This paper proposes a VAE for time-series data with two main contributions:
- an architecture design to approximately achieve "time-shift equivariance" defined in the paper;
- a training strategy which starts from splitting the data into shorter sequences and then grow the length of training sequences.

The experiments are pretty extensive in my opinion with many baselines and benchmark datasets, showing the competitiveness of the proposed method. Although one reviewer raised a question regarding outdated baselines.

I read the paper briefly, and while I can see the general idea of requiring time-shift equivariance as a condition to support the subsequent training strategy, I don't see what's the difference between the definition "time-shift equivariance" and stationarity (which is a well defined and studied concept in time series). Also the authors motivate the consideration in LSTM setting regarding "approximate equivariance", but there is no empirical validation regarding whether this "approximate equivariance" actually holds for the model before and after training. Why should one expect the converging behaviour of LSTM after long simulation? This should depend on the Eigenvalues of the recurrent matrices as well as the non-linearities? Even in such convergence case, I guess it means the hidden states have "died out" so that the generated time-series forecast will stay as constant (which is not what we want)?

I'd like to attach some post-rebuttal comments from the reviewers for the authors to consider:

- The main difficulty that I face with this paper is on the modelling/hypothesis side. The authors motivate the use of LSTM instead of transformer architectures since transformers are "less practical for very long sequences" (which is true due to the quadratic scale with the length). However, this confronts the main hypothesis of "approximate stationarity", since in those cases the statistical properties of the data are independent of time. Therefore, there should not be a need for long sequences in the first place. The answer on the posterior collapse of DVAEs with respect to the proposed architecture is also troubling. I do not understand how the authors extracted the conclusion that posterior collapse in their model is "not a defect but rather a natural consequence of the model’s design".

- The core concept of "time-shift equivariance" lacks a rigorous mathematical definition, and the reversion is even more confusing, as it transforms x from samples into a function while the paper further defines the probability p(x) of the function. The claim that "each sample can be generated with approximately the same probability starting at each time step" is also unconvincing, as the generation process clearly depends on context. The rationale for why the model design should satisfy time-shift equivariance is not clearly explained

**Resubmission Of Major Revision:**

The authors may consider submitting a major revision at a later time.